# PrkA controls peptidoglycan biosynthesis through the essential phosphorylation of ReoM

Sabrina Wamp[1], Zoe J Rutter[2], Jeanine Rismondo[1,3], Claire E Jennings[4], Lars Möller[5], Richard J Lewis[2†], Sven Halbedel[1]*

[1]FG11 - Division of Enteropathogenic bacteria and Legionella, Robert Koch Institute, Wernigerode, Germany; [2]Institute for Cell and Molecular Biosciences, Medical School, University of Newcastle, Newcastle upon Tyne, United Kingdom; [3]Department of General Microbiology, GZMB, Georg-August-Universität Göttingen, Göttingen, Germany; [4]Newcastle Drug Discovery, Northern Institute for Cancer Research, Newcastle upon Tyne, United Kingdom; [5]ZBS 4 - Advanced Light and Electron Microscopy, Robert Koch Institute, Berlin, Germany

**Abstract** Peptidoglycan (PG) is the main component of bacterial cell walls and the target for many antibiotics. PG biosynthesis is tightly coordinated with cell wall growth and turnover, and many of these control activities depend upon PASTA-domain containing eukaryotic-like serine/threonine protein kinases (PASTA-eSTK) that sense PG fragments. However, only a few PG biosynthetic enzymes are direct kinase substrates. Here, we identify the conserved ReoM protein as a novel PASTA-eSTK substrate in the Gram-positive pathogen *Listeria monocytogenes*. Our data show that the phosphorylation of ReoM is essential as it controls ClpCP-dependent proteolytic degradation of the essential enzyme MurA, which catalyses the first committed step in PG biosynthesis. We also identify ReoY as a second novel factor required for degradation of ClpCP substrates. Collectively, our data imply that the first committed step of PG biosynthesis is activated through control of ClpCP protease activity in response to signals of PG homeostasis imbalance.

*For correspondence:
halbedels@rki.de

Present address: †The Royal Society for the Protection of Birds, Sandy, United Kingdom

**Competing interests:** The authors declare that no competing interests exist.

## Introduction

The cell wall of Gram-positive bacteria is a complicated three-dimensional structure that engulfs the cell as a closed sacculus. The main component of bacterial cell walls is peptidoglycan (PG), a network of glycan strands crosslinked together by short peptides (*Vollmer et al., 2008a*). PG biosynthesis starts with the conversion of UDP-GlcNAc into lipid II, a disaccharide pentapeptide that is ligated to a membrane-embedded bactoprenol carrier lipid (*Typas et al., 2012*). This monomeric PG precursor is then flipped from the inner to the outer leaflet of the cytoplasmic membrane by MurJ- and Amj-like enzymes called flippases (*Ruiz, 2008*; *Sham et al., 2014*; *Meeske et al., 2015*). Glycosyltransferases belonging either to the bifunctional penicillin binding proteins (PBPs) or the SEDS (shape, elongation, division and sporulation) family then transfer the disaccharide pentapeptides to growing PG strands, which are finally crosslinked by a transpeptidation reaction catalysed by bifunctional (class A) or monofunctional (class B) PBPs (*Sauvage et al., 2008*; *Meeske et al., 2016*; *Emami et al., 2017*; *Taguchi et al., 2019*). Numerous hydrolytic or PG-modifying enzymes are also required to adapt the sacculus to the morphological changes that occur during bacterial cell growth and division (*Vollmer et al., 2008b*; *Uehara and Bernhardt, 2011*) or to alter its chemical properties for instance for immune evasion (*Moynihan et al., 2014*). A suite of regulators ensure that spatiotemporal control of PG synthesis is balanced against PG hydrolysis in cycles of bacterial growth and division (*Booth and Lewis, 2019*).

The activity of several key enzymes along the PG biosynthetic pathway is regulated by PASTA (PBP and serine/threonine kinase associated) domain-containing eukaryotic-like serine/threonine protein kinases (PASTA-eSTKs) (*Dworkin, 2015*; *Manuse et al., 2016*; *Egan et al., 2017*). These membrane-integral enzymes comprise a cytoplasmic kinase domain linked to several extracellular PASTA domains (*Manuse et al., 2016*). These proteins are stimulated by free muropeptides and lipid II (that accumulate during damage and turnover of PG) on interaction with their PASTA domains (*Mir et al., 2011*; *Hardt et al., 2017*; *Kaur et al., 2019*). PknB, a representative PASTA-eSTK from *Mycobacterium tuberculosis*, phosphorylates GlmU, a bifunctional uridyltransferase/acetyltransferase important for synthesis of UDP-GlcNAc, and in so doing reduces GlmU activity (*Parikh et al., 2009*). *M. tuberculosis* MviN, a MurJ-like flippase, is also a substrate of PknB and, in its phosphorylated state, P-MviN is inhibited by its binding partner, FhaA (*Gee et al., 2012*). *M. tuberculosis* PknB also phosphorylates both the class A PBP PonA1 (*Kieser et al., 2015*) and the amidase-like protein CwlM, which is essential for growth (*Deng et al., 2005*; *Boutte et al., 2016*; *Turapov et al., 2018*). CwlM is membrane-associated and interacts with MurJ to control lipid II export (*Turapov et al., 2018*). However, when phosphorylated, P-CwlM re-locates from the membrane to the cytoplasm (*Turapov et al., 2018*) where it allosterically activates MurA 20–40-fold (*Boutte et al., 2016*). MurA catalyzes the first committed step of PG biosynthesis by transferring an enoylpyruvate moiety to UDP-GlcNAc; MurA is essential in *M. tuberculosis* and in many other bacterial species tested (*Brown et al., 1995*; *Kock et al., 2004*; *Griffin et al., 2011*; *Rismondo et al., 2017*). Finally, the *Listeria monocytogenes* PASTA-eSTK, PrkA, phosphorylates YvcK, which is required for cell wall homeostasis in a so far unknown way (*Pensinger et al., 2016*).

Numerous additional proteins acting to coordinate cell wall biosynthesis with cell division are substrates of PASTA-eSTKs in other Gram-positive bacteria (*Manuse et al., 2016*), including the late cell division protein GpsB of *Bacillus subtilis* (*Macek et al., 2007*; *Pompeo et al., 2015*). We have shown previously that GpsB from *L. monocytogenes* is important for the last two steps of PG biosynthesis, *i. e.* transglycosylation and transpeptidation, by providing an assembly platform for the class A PBP, PBP A1 (*Rismondo et al., 2016*; *Cleverley et al., 2016*; *Cleverley et al., 2019*; *Halbedel and Lewis, 2019*), and this adaptor function of GpsB is maintained in at least *B. subtilis* and *Streptococcus pneumoniae* (*Cleverley et al., 2019*). An *L. monocytogenes* Δ*gpsB* mutant is impaired in PG biosynthesis and cannot grow at elevated temperatures (*Rismondo et al., 2016*), but this phenotype is readily corrected by a suppressor mutation, which mapped to *clpC* (*Rismondo et al., 2017*). ClpC is the ATPase subunit of the ClpCP protease that degrades substrate proteins upon heat stress (*Molière and Turgay, 2009*). MurA (*aka* MurAA in *B. subtilis*) is a ClpCP substrate in both *B. subtilis* and *L. monocytogenes* (*Kock et al., 2004*; *Rismondo et al., 2017*) and strongly accumulates in a *L. monocytogenes* Δ*clpC* mutant (*Rismondo et al., 2017*). Thus, a deficiency in the final two enzymatic steps of PG biosynthesis in the absence of GpsB is corrected by mutations in *clpC* that increase the amount of the first enzyme of the same PG biosynthetic pathway.

We here have isolated further *gpsB* suppressor mutations affecting previously unstudied *Listeria* genes. We demonstrate that these proteins control the ClpCP-dependent degradation of MurA in a PrkA-dependent and hitherto unprecedented manner. One of them is phosphorylated by PrkA and this phosphorylation is essential. Our results represent the first molecular link between PrkA-dependent protein phosphorylation and control of PG production in low G/C Gram-positive bacteria and explain how PG biosynthesis is adjusted in these bacteria to meet PG production and repair needs.

## Results

### *gpsB* suppressor mutations in the *lmo1503* (*reoM*) and *lmo1921* (*reoY*) genes

A *L. monocytogenes* Δ*gpsB* mutant is unable to replicate at 42°C, but readily forms suppressors correcting this defect (*Rismondo et al., 2017*). Previously isolated *gpsB* suppressors carried a mutation in the *clpC* gene, important for the stability of the UDP-*N*-acetylglucosamine 1-carboxyvinyltransferase MurA (*Rismondo et al., 2017*). We have characterised three more *shg* (suppression of heat sensitive growth) suppressor mutants (*shg8*, *shg10* and *shg12*) isolated from a Δ*gpsB* mutant incubated on a BHI agar plate at 42°C. These three *shg* strains grew as fast as the wild type when cultivated at

37°C or 42°C, whereas the parental Δ*gpsB* mutant grew at a reduced rate at 37°C and did not grow at 42°C (*Figure 1A–B*), as shown previously (*Rismondo et al., 2016*).

Sequencing of the *shg8*, *shg10* and *shg12* genomes identified one SNP in each strain that was absent from the parental Δ*gpsB* mutant. Strain *shg8* carried a mutation in the uncharacterized *lmo1921* gene (herein named *reoY*, see below) that exchanged H87 into tyrosine; the same gene was affected by the introduction of a premature stop codon after the 73$^{rd}$ *reoY* codon in strain *shg10*. Strain *shg12* carried a mutation in the ribosomal binding site (RBS) of the *lmo1503* gene

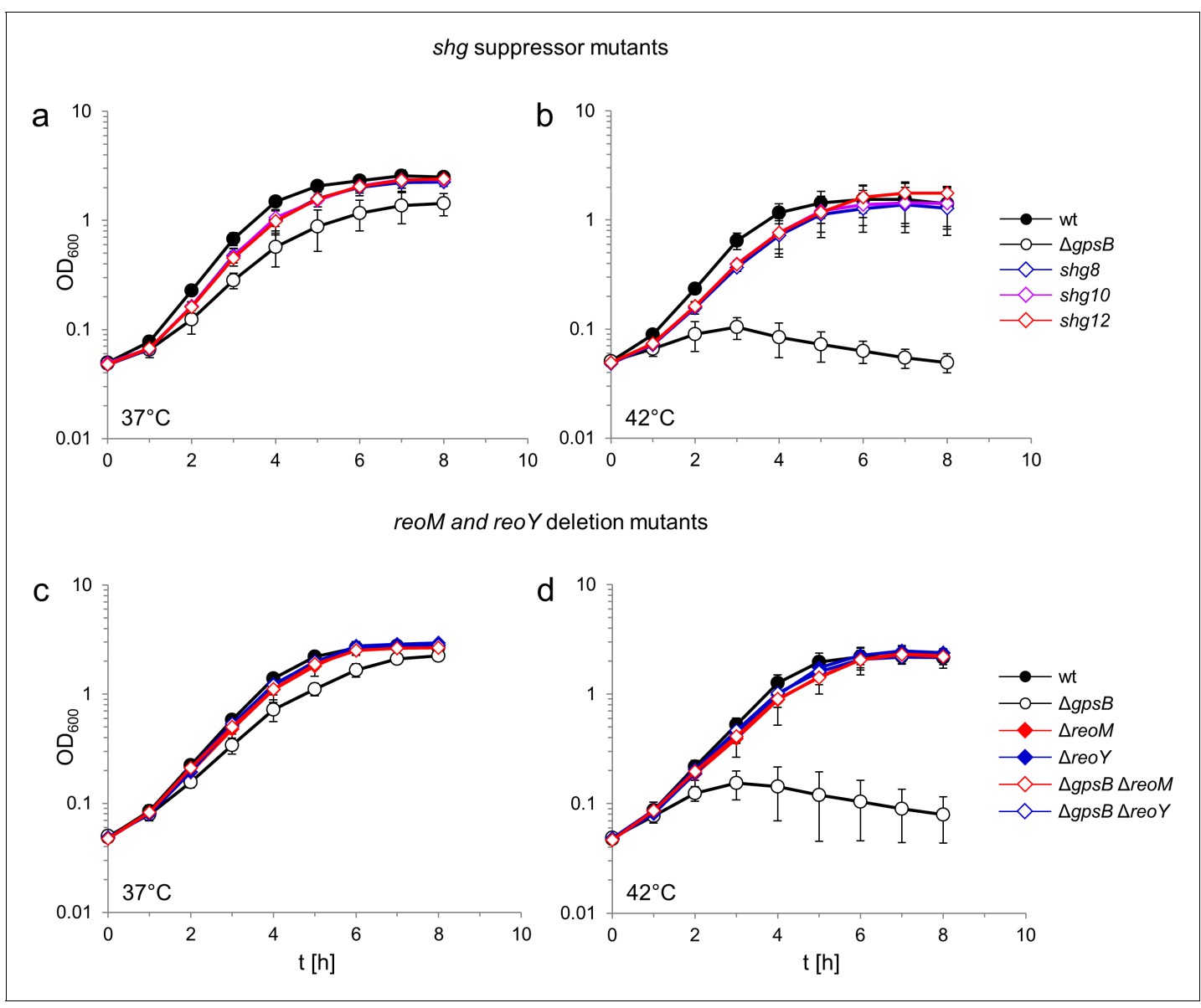

**Figure 1.** Suppression of the growth defects of a *L. monocytogenes* Δ*gpsB* mutant by *reoM* and *reoY* mutations. (A–B) Effect of suppressor mutations on growth of the Δ*gpsB* mutant. Growth of *L. monocytogenes* strains EGD-e (wt), LMJR19 (Δ*gpsB*), *shg8* (Δ*gpsB reoY* H87Y), *shg10* (Δ*gpsB reoY* TAA74) and *shg12* (Δ*gpsB reoM* RBS mutation) in BHI broth at 37°C (A) and 42°C (B). (C–D) Effect of Δ*reoM* and Δ*reoY* deletions on growth of *L. monocytogenes*. Growth of *L. monocytogenes* strains EGD-e (wt), LMJR19 (Δ*gpsB*), LMSW30 (Δ*reoM*), LMSW32 (Δ*reoY*), LMJR137 (Δ*gpsB* Δ*reoM*) and LMJR120 (Δ*gpsB* Δ*reoY*) in BHI broth was recorded at 37°C (C) and 42°C (D). All growth experiments were performed three times and average values and standard deviations are shown.

The online version of this article includes the following figure supplement(s) for figure 1:

**Figure supplement 1.** Overexpression of *reoM* but not *reoY* affects growth of the *L. monocytogenes* Δ*gpsB* mutant.
**Figure supplement 2.** Effect of *reoM* and *reoY* deletions on cell morphology.

(renamed *reoM*), encoding an IreB-like protein, the function of which is not understood (*Hall et al., 2013*).

Whether the mutation in the RBS of *reoM* in strain *shg12* affected *reoM* expression was not clear. Therefore, the *reoM* gene was deleted from the genome of the wild type and the Δ*gpsB* mutant. While deletion of *reoM* had no effect on growth of wild type bacteria, it completely suppressed the growth defects of the Δ*gpsB* mutant at both 37°C and 42°C (*Figure 1C–D*). It is thus likely that the mutation in the *reoM* RBS impairs its expression. Likewise, deletion of *reoY* completely restored growth of the Δ*gpsB* mutant at both temperatures (*Figure 1C–D*).

Expression of an additional copy of *reoM* impaired growth of the Δ*gpsB* mutant without affecting the growth of wild type bacteria, whilst expression of a second copy of *reoY* had no effect (*Figure 1—figure supplement 1A,B*). The expression of *reoM* is thus inversely correlated with the growth of the Δ*gpsB* mutant. Finally, the physiology of the Δ*reoM* and Δ*reoY* mutants was examined; their cell lengths were wild type-like and unaffected by the presence or absence of *gpsB*, suggesting the absence of cell division defects in the Δ*reoM* or Δ*reoY* mutants (*Figure 1—figure supplement 2A,B*). Scanning electron micrographs of Δ*reoM* and Δ*reoY* single mutants revealed that these bacteria had a normal rod-shape, but that the Δ*gpsB* Δ*reoM* and Δ*gpsB* Δ*reoY* double mutants were partially bent (*Figure 1—figure supplement 2C*), implying the presence of some shape maintenance defects along the lateral cell cylinders.

## ReoM and ReoY affect the stability of MurA

Suppression of the Δ*gpsB* phenotype can be achieved by the accumulation of MurA (*Rismondo et al., 2017*). Consequently, MurA levels were determined in Δ*reoM* and Δ*reoY* mutant strains by western blotting. MurA accumulated by at least eight-fold in comparison to the wild type in the absence of *reoM* or *reoY* (*Figure 2A*), and reached similar levels to a mutant lacking *clpC*, which encodes the ATPase subunit of the ClpCP protease (*Figure 2A*). MurAA, the *B. subtilis* MurA homologue, is degraded by the ClpCP protease in vivo (*Kock et al., 2004*). In order to test whether *reoM* and *reoY* exert their effect on MurA in a ClpC-dependent manner in *L. monocytogenes*, MurA levels were determined in Δ*clpC* Δ*reoM* and Δ*clpC* Δ*reoY* double mutants. The MurA levels in Δ*clpC*, Δ*reoM* and Δ*reoY* single mutants were the same as in Δ*clpC* Δ*reoM* and Δ*clpC* Δ*reoY* double mutant strains (*Figure 2B*). Likewise, the MurA level in a mutant lacking *murZ*, previously shown to contribute to MurA accumulation (*Rismondo et al., 2017*), is not additive to the MurA level in Δ*clpC* cells (*Figure 2B*). Reintroduction of *reoM*, *reoY* and *murZ* into their respective single mutant backgrounds complemented their phenotypes (*Figure 2—figure supplement 1A* and Figure 5C below). Therefore, ReoM, ReoY and MurZ likely affect the ClpCP-dependent degradation of MurA. Combinations of Δ*reoM*, Δ*reoY* and Δ*murZ* deletions did also not exert any additive effect on accumulation of MurA (*Figure 2—figure supplement 1B*), further validating the conclusion that these genes all belong to the same pathway.

We then tested the hypothesis that ReoM and ReoY control proteolytic stability of MurA and followed MurA and DivIVA degradation over time in cells that had been treated with chloramphenicol to block protein biosynthesis. MurA was almost completely degraded in wild type cells 80 min after chloramphenicol treatment (*Figure 2C*), whereas DivIVA was stable (*Figure 2—figure supplement 2B*). By contrast, no MurA degradation was observed in mutants lacking *clpC*, *reoM* or *reoY* (*Figure 2C*), which together demonstrates that ReoM and ReoY are as important for MurA degradation as is ClpC.

## The effect of ReoM and ReoY on MurA levels is conserved

Homologues of the 90-residue ReoM protein are found across the entire Firmicute phylum, and include IreB, a substrate of the protein serine/threonine kinase IreK and its cognate phosphatase IreP from *Enterococcus faecalis* (*Hall et al., 2013*), whereas ReoY homologues are present only in the *Bacilli*. A *reoY* homologue has been identified as a Δ*ireK* suppressor in *E. faecalis* (*Banla et al., 2018*), but the function of the *E. faecalis reoY* and *reoM* homologues remains unknown. In *B. subtilis*, ReoM corresponds to YrzL (e-value 3e$^{-29}$) and ReoY to YpiB (4e$^{-61}$), but neither protein has been studied thus far. To assess whether YrzL and YpiB were also crucial for control of MurAA levels in *B. subtilis*, cellular protein extracts from *B. subtilis* Δ*yrzL* and Δ*ypiB* mutants were probed by western blot (*Figure 2D*). MurAA accumulated by at least 12-fold in these strains in comparison to the wild

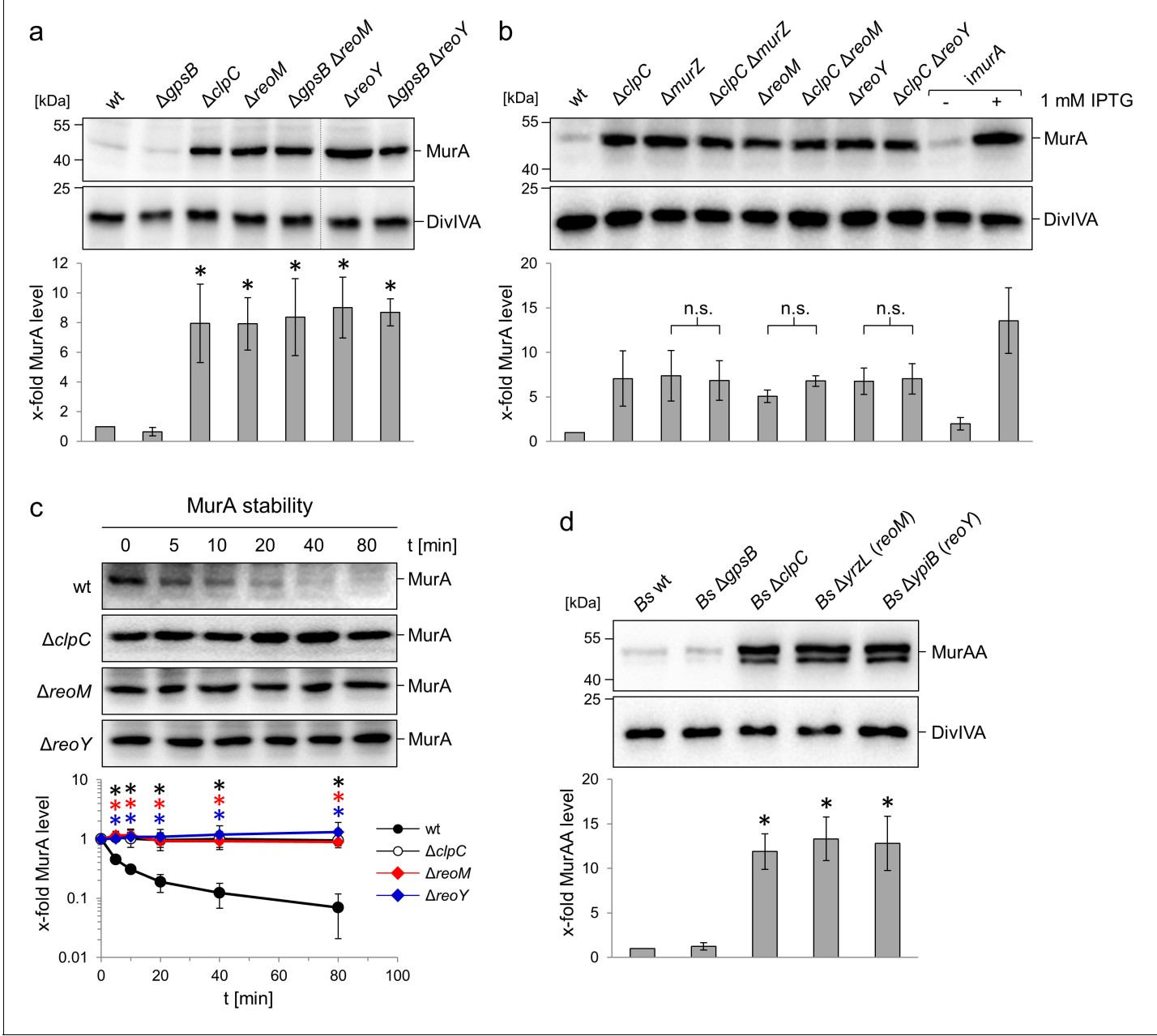

**Figure 2.** Effect of the *reoM*, *reoY* and *clpC* genes on levels of MurA in *L. monocytogenes* and MurAA in *B. subtilis*. (A) Effect of *reoM* and *reoY* deletions (single or when combined with *gpsB* deletion) on MurA (above) and DivIVA levels (middle) in *L. monocytogenes* strains EGD-e (wt), LMJR19 (Δ*gpsB*), LMSW30 (Δ*reoM*), LMSW32 (Δ*reoY*), LMJR137 (Δ*gpsB* Δ*reoM*) and LMJR120 (Δ*gpsB* Δ*reoY*) and quantification of MurA levels (below). Strain LMJR138 (Δ*clpC*) was included for comparison. Non-relevant lanes were excised from the blots (dotted lines). Average values ± standard deviations were shown (n = 3). Statistically significant differences compared to wild type are marked by asterisks (p<0.05, *t*-test). (B) Effect of *reoM*, *reoY* and *murZ* deletions when combined with *clpC* deletion on MurA (above) and DivIVA levels (middle) in *L. monocytogenes* strains EGD-e (wt), LMJR138 (Δ*clpC*), LMJR104 (Δ*murZ*), LMJR171 (Δ*clpC* Δ*murZ*), LMSW30 (Δ*reoM*), LMSW50 (Δ*clpC* Δ*reoM*), LMSW32 (Δ*reoY*) and LMSW51 (Δ*clpC* Δ*reoY*) and quantification of MurA levels (below). Strain LMJR123 (*imurA*, i - is used to denote IPTG-dependent alleles) grown in the presence or absence of IPTG was included for comparison. Average values and standard deviations were shown (n = 3) and n. s. means not significant (p<0.05, *t*-test). (C) Western blots following MurA degradation in vivo. *L. monocytogenes* strains EGD-e (wt), LMJR138 (Δ*clpC*), LMSW30 (Δ*reoM*) and LMSW32 (Δ*reoY*) were grown to an OD$_{600}$ of 1.0 and 100 µg/ml chloramphenicol was added. Samples were taken before chloramphenicol addition and after several time intervals to analyse MurA levels. MurA signals were quantified by densitometry and average values and standard deviations are shown (n = 3). Statistically significant differences are marked with asterisks (p<0.05, *t*-test). (D) Effect of the *reoM* and *reoY* homologues *yrzL* and *ypiB*, respectively, on MurAA (above) and DivIVA levels (middle) of *B. subtilis* and quantification of MurAA levels (below). Strains BKE00860 (Δ*clpC*), BKE22180 (Δ*gpsB*), BKE22580 (Δ*ypiB*/*reoY*) and BKE27400 (Δ*yrzL*/*reoM*) were grown to mid-logarithmic growth phase before total cellular proteins were isolated. *B. subtilis* 168 (wt) was included as control. That

*Figure 2 continued on next page*

*Figure 2 continued*

MurAA is detected in two isoforms had been observed earlier but the reasons for this are not known (*Kock et al., 2004*). Average values and standard deviations were shown (n = 3). Asterisks indicate statistically significant differences compared to wild type (p<0.05, *t*-test).

The online version of this article includes the following figure supplement(s) for figure 2:

**Figure supplement 1.** Complementation and epistasis experiments.
**Figure supplement 2.** DivIVA stability in *L. monocytogenes* Δ*clpC*, Δ*reoM* and Δ*reoY* mutants.
**Figure supplement 3.** Effect of *reoM* and *reoY* deletions on accumulation of other ClpC substrates in *B. subtilis*.

type. Furthermore, the amount of MurAA was also increased by 12-fold in the Δ*clpC* mutant. Taken together, these data indicate that ReoM and ReoY functions are conserved in both species. We thus propose to rename *lmo1503* (*yrzL*) as *reoM* (regulator of MurA(A) degradation) and analogously *lmo1921* (*ypiB*) as *reoY*.

Several other ClpC substrates are known in *B. subtilis*, including the glutamine fructose-6-phosphate transaminase GlmS and the acetolactate synthase subunit IlvB (*Gerth et al., 2008*). The levels of both proteins were also significantly increased in *B. subtilis* Δ*reoM* and Δ*reoY* mutants (*Figure 2—figure supplement 3*), indicating that ReoM and ReoY are required for degradation of ClpC substrates in general.

## ReoM and ReoY contribute to PG biosynthesis

In order to test whether MurA accumulation affected PG production, we tested the effect of enhanced MurA levels on resistance of *L. monocytogenes* against the cephalosporin antibiotic ceftriaxone. Artificial overproduction of MurA in strain LMJR116, which carries an IPTG-inducible *murA* gene in addition to the native copy on the chromosome, lead to a 12-fold increase of ceftriaxone resistance, while MurA depletion lowered ceftriaxone resistance (*Figure 3A*). MurA levels are thus directly correlated with PG production, presumably leading to stimulation or impairment of PG biosynthesis during overproduction and depletion, respectively. In good agreement with the overproduction of MurA, ceftriaxone resistance of the Δ*clpC* mutant increased to the same degree as when MurA was overproduced (*Figure 3A*). Ceftriaxone resistance of Δ*reoM*, Δ*reoY* and Δ*murZ* mutants increased two- to three-fold (*Figure 3A*); this intermediate resistance level is probably explained by the presence of functional ClpCP in these strains. Nevertheless, these observations are consistent with a function of ReoM, ReoY and MurZ as regulators of ClpCP-dependent MurA degradation. In good agreement with this concept of stimulated PG biosynthesis during MurA accumulation, we observed thicker PG layers at the cell poles of Δ*reoM* and Δ*reoY* mutants, which also have more uneven PG layers along their lateral wall, whereas both phenomena were not observed in wild type cells (*Figure 3B*; *Figure 3—figure supplement 1*). Moreover, Δ*reoM*, Δ*reoY* and Δ*murZ* mutants showed salt-sensitive growth (*Figure 3C*), which is a known phenotype of the *L. monocytogenes* Δ*clpC* mutant (*Rouquette et al., 1996*). Salt sensitivity of the Δ*reoM* mutant was as severe as for the Δ*clpC* mutant, whereas the Δ*reoY* and Δ*murZ* mutants showed milder phenotypes (*Figure 3C*). Taken together, these results indicate that modulation of MurA levels effectively controls PG biosynthesis and also demonstrate that ReoM, ReoY and MurZ play an important role in its regulation.

## Phosphorylation and dephosphorylation of ReoM by PrkA and PrpC in vitro

PrkA (encoded by *lmo1820*) and PrpC (*lmo1821*) are the *L. monocytogenes* homologues of *E. faecalis* IreK and IreP, respectively. Consequently, the pairwise interactions and biochemical properties of ReoM, the PrkA kinase domain (PrkA-KD) and the cognate phosphatase PrpC were investigated. All isolated proteins electrophoresed as single species in non-denaturing PAGE (lanes 1, 2, *Figure 4A*; lanes 1–4, *Figure 4B*). When ReoM was incubated with PrkA-KD, in the absence of ATP, a slower migrating species was observed and the individual bands corresponding to ReoM and PrkA-KD disappeared indicating that the slower migrating species was a ReoM:PrkA-KD complex (lane 3, *Figure 4A*). When ReoM was incubated with PrkA-KD and Mg/ATP under the same conditions, free PrkA-KD was observed but no bands equivalent to ReoM and the ReoM:PrkA-KD complex remained; instead a new species was present, migrating faster in the gel than ReoM (lane 4, *Figure 4A*), which is likely to be phosphorylated ReoM (P-ReoM). Intact protein liquid chromatography-mass

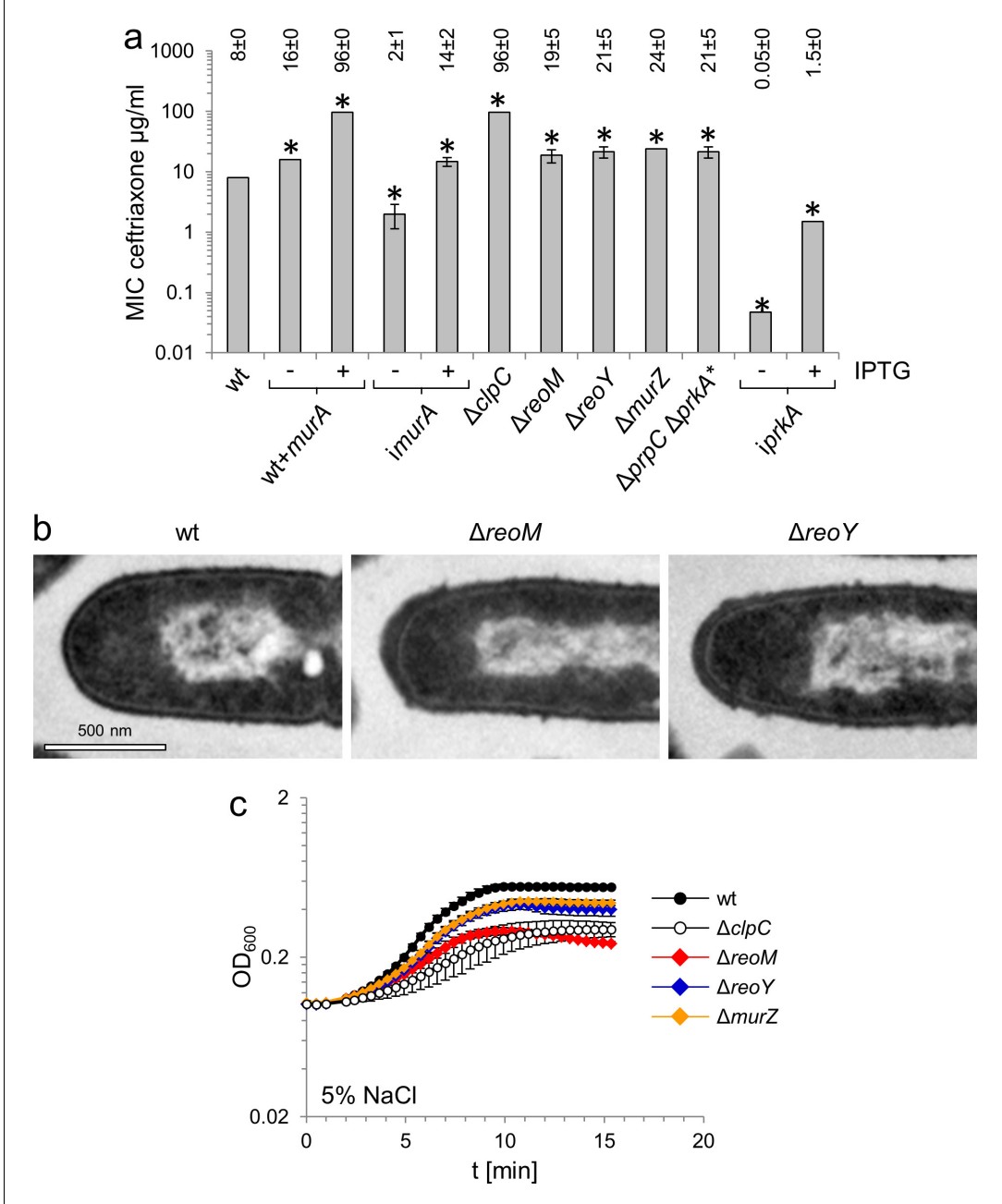

**Figure 3.** MurA accumulation affects peptidoglycan biosynthesis and salt sensitivity. (**A**) Minimal inhibitory concentrations (MIC) for ceftriaxone of mutants with altered MurA accumulation. Average values and standard deviations are calculated from three independent experiments and given above the panel. Asterisks indicate statistically significant differences compared to wild type (p<0.05, *t*-test). Please note that the i*prkA* strain showed residual growth on BHI agar plates not containing IPTG, even though it required IPTG for growth in BHI broth. (**B**) Transmission electron microscopy of ultrathin sections of fixed whole cells of *L. monocytogenes* wildtype, Δ*reoM* and Δ*reoY* mutants. *L. monocytogenes* strains EGD-e (wt), LMSW30 (Δ*reoM*) and LMSW32 (Δ*reoY*) were grown to mid-logarithmic growth phase in BHI broth at 37°C and subjected to chemical fixation and subsequent electron microscopy as described in the experimental procedures section. (**C**) Salt sensitive growth of mutants with altered MurA accumulation. *L. monocytogenes* strains EGD-e (wt), LMJR138 (Δ*clpC*), LMSW30 (Δ*reoM*), LMSW32 (Δ*reoY*) and LMJR104 (Δ*murZ*) were grown in BHI broth containing 5% NaCl at 37°C. Average values and standard deviations are calculated from three independent experiments.

The online version of this article includes the following figure supplement(s) for figure 3:

**Figure supplement 1.** ReoM and ReoY affect thickness of polar peptidoglycan.

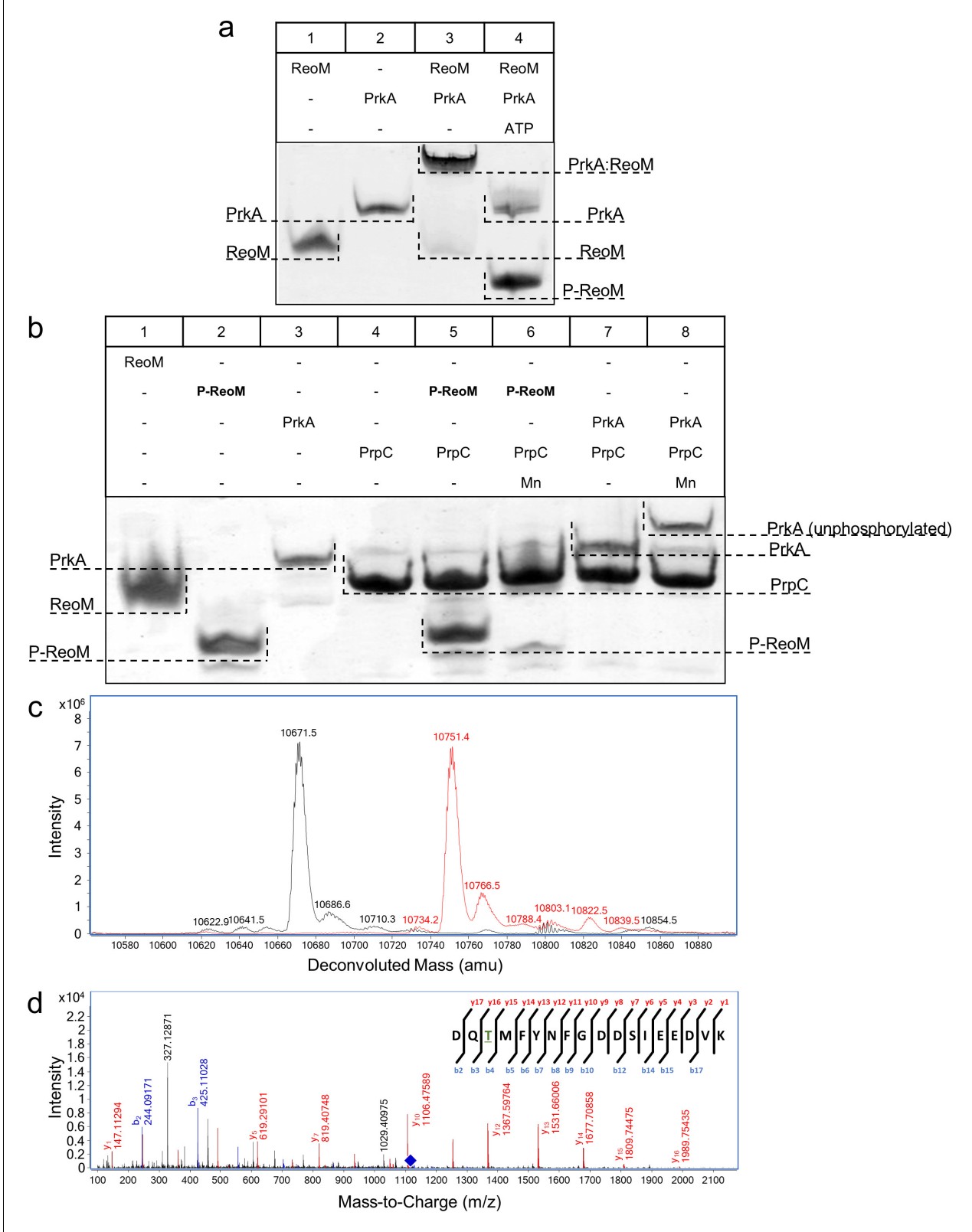

**Figure 4.** The PrkA/PrpC pair controls the phosphorylation status of ReoM. (A–B) Non-denaturing, native PAGE analysis of the phosphorylation (A) and dephosphorylation (B) of ReoM in vitro. The components of each lane in the Coomassie-stained gel are annotated above the image and the position and identity of relevant bands is marked to the side. (C) LC-MS analysis of intact ReoM. The deconvoluted mass spectrum for non-phosphorylated ReoM (black) is overlaid over the equivalent spectrum for mono-phosphorylated ReoM, P-ReoM (red). (D) LC-MS/MS was used to perform peptide

*Figure 4 continued on next page*

*Figure 4 continued*

mapping analysis that revealed that Thr7 is the sole phosphosite of ReoM. The MS/MS fragmentation spectra of the phosphorylated peptide encompassing Asp5-Lys22 is presented with *b*-ion fragmentation in blue and *y*-ion fragmentation shown in red, whilst the precursor ion (m/z 1116.86, z = 2+) is represented by a blue diamond.

The online version of this article includes the following figure supplement(s) for figure 4:

**Figure supplement 1.** LC-MS analysis of intact ReoM.
**Figure supplement 2.** LC-MS analysis of ReoM T7A.
**Figure supplement 3.** Dephosphorylation of P-ReoM by PrpC.
**Figure supplement 4.** Dephosphorylation of P-PrkA-KD by PrpC.

spectrometry (LC-MS) analysis of ReoM isolated from PrkA-KD after phosphorylation revealed the addition of 79.9 Da in comparison to the mass of ReoM (10671.5 Da), which corresponds to the formation of a singly-phosphorylated ReoM product of 10751.4 Da (*Figure 4C*, *Figure 4—figure supplement 1*). MS/MS spectra obtained during peptide mass fingerprinting were also consistent with one phosphorylation event per protein chain; one ReoM peptide, spanning residues Asp5 to Lys22 with mass of 2151.89 Da, was increased by 79.96 Da after incubation with PrkA-KD and Mg/ATP. Analysis of the *y*- and *b*- ions in the MS/MS fragmentation spectrum of this peptide was consistent only with Thr7 as the sole phosphosite in ReoM (*Figure 4D*). Finally, mutation of Thr7 to alanine completely abrogated the phosphorylation of ReoM by PrkA-KD when analysed by LC-MS (*Figure 4—figure supplement 2*).

The ability of PrpC, the partner phosphatase to PrkA in *L. monocytogenes,* to interact with and remove phosphoryl groups from PrkA-KD and P-ReoM was also tested in vitro. PrkA and purified P-ReoM were each incubated with PrpC in the absence and presence of $MnCl_2$, since divalent cations are essential co-factors for the PPM phosphatase family to which PrpC belongs (*Kennelly, 2001*), and the products were analysed by non-denaturing PAGE. Unlike the situation with ReoM and PrkA-KD, no stable protein:protein complexes were formed either in the presence or absence of endogenous $MnCl_2$ (*Figure 4B*). The incubation of P-ReoM with PrpC and manganese resulted in the almost complete disappearance of the band corresponding to P-ReoM (lane 6, *Figure 4B*) in comparison to the same reaction conducted without the addition of $MnCl_2$ (lane 5, *Figure 4B*). The new band, corresponding to ReoM alone in lane 6, is masked by that for PrpC that migrates similarly to ReoM (lanes 1 and 4, *Figure 4B*) under these electrophoresis conditions. The presence of unphosphorylated ReoM and the absence of P-ReoM was confirmed by LC-MS (*Figure 4—figure supplement 3*). When incubated with PrpC in the presence of manganese ions, the band for PrkA-KD electrophoresed more slowly than for PrkA-KD in isolation (lanes 3 and 8, *Figure 4B*), indicating that PrkA-KD had been dephosphorylated by PrpC. LC-MS analysis of PrkA-KD that had been incubated with PrpC/$MnCl_2$ yielded a single major species of 37,413.2 Da, consistent with the predicted mass of the expressed recombinant construct, and the absence of a peak corresponding to phosphorylated PrkA-KD, P-PrkA-KD (*Figure 4—figure supplement 4*). Therefore, PrkA-KD is capable of autophosphorylation even when expressed in a heterologous host, consistent with previous observations made for similar PASTA-eSTKs from other Gram-positive bacteria (*Madec et al., 2003*; *Kristich et al., 2011*). Finally, in the absence of $MnCl_2$ no change in electrophoretic mobility was observed for P-PrkA-KD (lane 7, *Figure 4B*).

## Phosphorylation of ReoM at threonine seven is essential for viability

PrkA phosphorylates ReoM on Thr7 and PrpC reverses this reaction in vitro; ReoM phosphorylation at Thr7 in vivo has also been observed by phosphoproteomics (*Misra et al., 2011*). In the absence of molecular details on the impact of Thr7 phosphorylation we determined the importance of this phosphorylation in vivo by engineering a phospho-ablative T7A exchange in an IPTG-inducible allele of *reoM* and introduced it into the Δ*reoM* mutant background. Deletion, depletion or expression of wildtype *reoM* had no effect on growth in strains LMSW30 (Δ*reoM*) and LMSW57 (i*reoM*, i - is used to denote IPTG-dependent alleles throughout the manuscript) at 37˚C. Likewise, strain LMSW52 (i*reoM T7A*) grew normally in the absence of IPTG. However, the *reoM* mutant with the T7A mutation did not grow at all in the presence of IPTG, when expression of the phospho-ablative *reoM T7A* allele was induced (*Figure 5A*), suggesting that phosphorylation of ReoM at Thr7 is essential for the

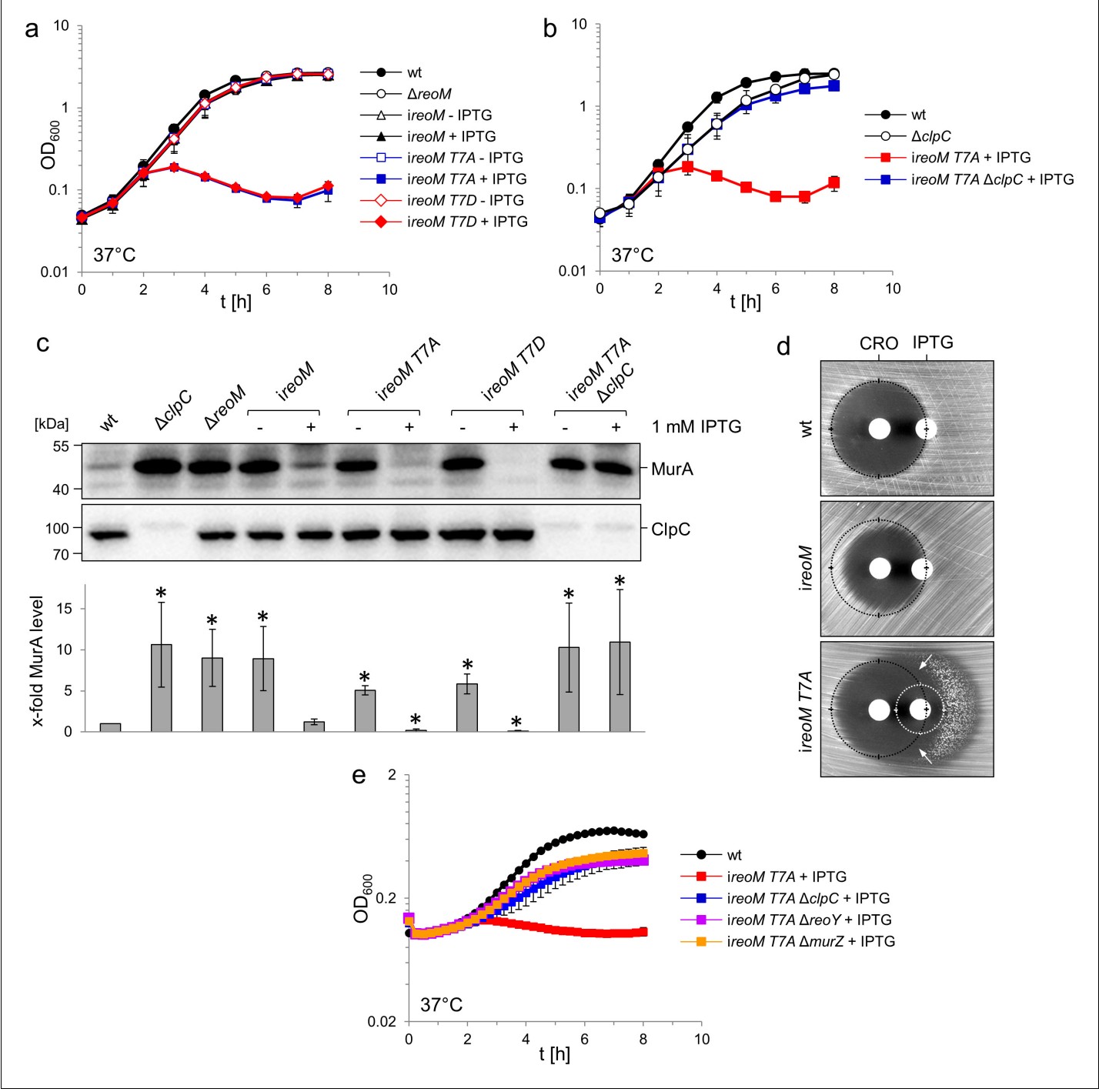

**Figure 5.** A ReoM T7A exchange affects growth and MurA levels in a ClpC-dependent manner. (**A**) Lethality of the *reoM T7A* and *reoM T7D* mutations in *L. monocytogenes*. *L. monocytogenes* strains EGD-e (wt), LMSW30 (Δ*reoM*), LMSW57 (i*reoM*), LMSW52 (i*reoM T7A*) and LMSW53 (i*reoM T7D*) were grown in BHI broth ±1 mM IPTG at 37°C. The experiment was repeated three times and average values and standard deviations are shown. (**B**) Suppression of *reoM T7A* lethality by deletion of *clpC*. *L. monocytogenes* strains EGD-e (wt), LMJR138 (Δ*clpC*), LMSW52 (i*reoM T7A*) and LMSW72 (i*reoM T7A* Δ*clpC*) were grown in BHI broth ±1 mM IPTG at 37°C. The experiment was repeated three times and average values and standard deviations are shown. (**C**) Western blot showing cellular levels of MurA (top) and ClpC (middle) in the strains included in panels A and B. For this experiment, strains were grown in BHI broth not containing IPTG at 37°C. IPTG (1 mM) was added to the cultures at an OD$_{600}$ of 0.2 and the cells were collected 2 hr later. Quantification of MurA signals by densitometry is shown below the western blots. Average values and standard deviations calculated from three independent experiments are shown. Asterisks indicate statistically significant differences (p<0.05, *t*-test). (**D**) ReoM T7A expression sensitises *L. monocytogenes* against ceftriaxone. Synergism between ceftriaxone and IPTG in the i*reoM T7A* strain LMSW52 in a disc

*Figure 5 continued on next page*

Figure 5 continued

diffusion assay with filter discs containing 50 mg/ml ceftriaxone (CRO, left) and 1 mM IPTG (right). For comparison, wild type levels of growth inhibition by ceftriaxone are marked with black circles. Zone of growth inhibition by IPTG in the i*reoM T7A* mutant is marked with a white circle. Please note that strain LMSW52 shows hetero-resistance against IPTG (two zones of growth inhibition with different resistance levels). Arrows mark the zones of synergism between ceftriaxone and IPTG. (E) Contribution of ReoY and MurZ to the lethal *reoM T7A* phenotype. *L. monocytogenes* strains EGD-e (wt), LMSW52 (i*reoM T7A*), LMSW72 (i*reoM T7A ΔclpC*), LMSW123 (i*reoM T7A ΔreoY*) and LMSW124 (i*reoM T7A ΔmurZ*) were grown in BHI broth containing 1 mM IPTG and growth at 37°C was recorded in a microplate reader. Average values and standard deviations were calculated from an experiment performed in triplicate.

viability of *L. monocytogenes*. We next engineered a *reoM T7D* mutant to mimic the effect of Thr7 phosphorylation. However, the resulting strain was as sensitive to IPTG as the *reoM T7A* mutant (*Figure 5A*). Since ReoM influences the proteolytic stability of MurA, we determined the cellular amount of MurA in strains expressing the T7A/T7D variants of ReoM. For this purpose, strains LMSW57 (i*reoM*), LMSW52 (i*reoM T7A*) and LMSW53 (i*reoM T7D*) were initially cultivated in plain BHI broth. At an $OD_{600}$ of 0.2, IPTG was added to a final concentration of 1 mM and cells were harvested 2 hr later. Strain LMSW57 (i*reoM*) showed Δ*clpC*-like MurA accumulation (around seven-fold in this experiment) when cultured in the absence of IPTG, but MurA was present at wild type levels in the presence of IPTG (*Figure 5C*). The strains with the T7A and T7D exchanges also accumulated MurA to a Δ*clpC*-like extent in the absence of IPTG. However, only a minor fraction of the wild type MurA levels could be detected in cells expressing the *reoM T7A* (17 ± 2%) or *reoM T7D* alleles (10 ± 2%, *Figure 5C*). That the *reoM T7D* mutant does not have the opposite phenotype as the *reoM T7A* mutant indicates that ReoM T7D behaves as a non-phosphorylatable protein and not as a genuine phospho-mimetic variant. The reasons for this discrepancy are currently not clear, but phospho-mimetic mutations do not work in all cases (*Dephoure et al., 2013*), since aspartate (and glutamate) are unfaithful chemical mimics of phosphothreonine; a similar phenomenon was also observed with phospho-mimetic replacements of Thr7 in *E. faecalis* IreB (*Hall et al., 2013*). Nonetheless, our data demonstrate that Thr7 in ReoM is of special importance for the proteolytic stability of MurA. In agreement with these results, IPTG was toxic for the i*reoM T7A* mutant in a disc diffusion assay and rendered this strain hypersensitive to ceftriaxone (*Figure 5D*).

## Lethality of the *reoM T7A* mutations depends on ClpC

That MurA is rapidly degraded in cells expressing *reoM T7A* implies that phosphorylation/dephosphorylation of ReoM at Thr7 controls ClpCP-dependent MurA degradation. MurA is an essential enzyme in *L. monocytogenes* (*Rismondo et al., 2017*), and stimulation of ClpCP-dependent MurA degradation in the *reoM T7A* mutant would provide an explanation for the lethality of this mutation. In order to address this possibility, we deleted *clpC* in the conditional i*reoM T7A* background. This strain grew even in the presence of IPTG, a compelling demonstration that the removal of *clpC* suppressed the lethality of the *reoM T7A* mutation (*Figure 5B*). MurA also accumulated to the same degree as in the Δ*clpC* mutant in this strain (*Figure 5C*), suggesting that inactivation of the ClpCP-dependent degradation of MurA overcame the lethal effect of the T7A mutation in *reoM* and this suggests that ClpCP acts downstream of ReoM. We next wondered whether deletion of *reoY* and *murZ* would have a similar effect and deleted these genes in the *reoM T7A* mutant. As can be seen in *Figure 5E*, deletion of either gene overcame the lethality of *reoM T7A*, indicating that ReoY and MurZ must also act downstream of ReoM.

## Crystal structure of ReoM, a homologue of *Enterococcus faecalis* IreB

Purified ReoM yielded crystals that diffracted to a maximum resolution of 1.6 Å. The NMR structure of IreB (PDBid 5US5) (*Hall et al., 2017*) was used to solve the structure of ReoM by molecular replacement (*Figure 6A*). The data collection and refinement statistics for the ReoM structure are summarised in *Table 1*. ReoM shares the same overall fold as IreB (*Hall et al., 2017*), each containing a compact 5-helical bundle (four standard α-helices and one single-turned $3_{10}$-helix between residues 52 and 54) with short loops between the secondary structure elements, which are defined above the sequence alignment in *Figure 6B*. Other than IreB (*Hall et al., 2017*), there are no structural homologues of ReoM with functional significance in the PDB. The helical bundles in both ReoM and IreB associate into homodimers with α-helices two and four from each protomer forming the

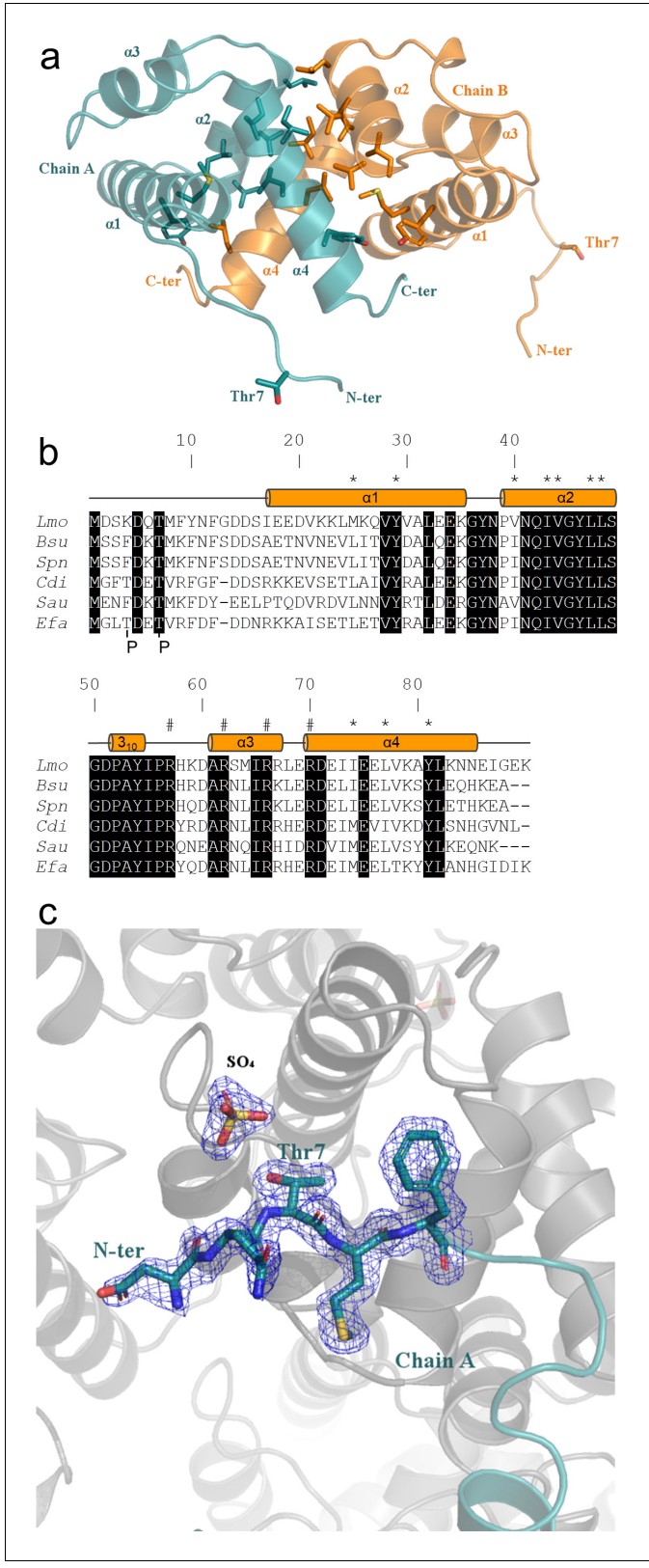

**Figure 6.** Crystal structure of ReoM. (**A**) The structure of ReoM depicted as a cartoon with each protomer in the dimer coloured separately (cyan and orange). The secondary structure elements are numbered according to their position in the amino acid sequence. Thr7 and some of the key amino acids in the dimer interface and the hydrophobic core are drawn as stick figures. (**B**) Sequence alignment of ReoM (*Lmo*) and its homologues from

*Figure 6 continued on next page*

*Figure 6 continued*

*Bacillus subtilis* (*Bsu*), *Streptococcus pneumoniae* (*Spn*), *Clostridium difficile* (*Cdi*) and *Staphylococcus aureus* (*Sau*) with the sequence of IreB from *Enterococcus faecalis* (*Efa*) underneath. Amino acid sequence numbers pertain to ReoM and the site of phosphorylation in ReoM (Thr7) and the twin phosphorylations in IreB (minor site: Thr4; major site: Thr7) are highlighted. Invariant amino acids are shaded black, residues in the ReoM dimer interface have an asterisk above, and the secondary structure elements are defined by cylinders above the alignment. Arginine residues mutated in this study are indicated by a hashtag above the alignment. (C) The final 2F$_{obs}$-F$_{calc}$ electron density map, contoured at a level of 0.42 e$^-$/Å$^3$, of the N-terminal region in the immediate vicinity of Thr7 in chain A of the ReoM dimer indicates that the protein model could be built with confidence even though this region contains no secondary structure elements.

The online version of this article includes the following figure supplement(s) for figure 6:

**Figure supplement 1.** ReoM and P-ReoM have the same oligomeric state.

**Figure supplement 2.** Lethality of ReoM R57A and R62A substitutions.

**Figure supplement 3.** A possible conformational change of the flexible ReoM N-terminus induced by phosphorylation.

**Table 1.** Summary of the data collection and refinement statistics for ReoM.

**Data collection**

| | |
|---|---|
| Beamline | Diamond I03 |
| Wavelength (Å) | 0.976 |
| Resolution (Å) | 74.45–1.60 (1.63–1.60)* |
| Space group | P 2$_1$ 2$_1$ 2$_1$ |
| *a, b, c* (Å) | 38.79, 58.62, 74.45 |
| α, β, γ (°) | 90, 90, 90 |
| R$_{pim}$ | 0.064 (0.533) |
| CC (1/2) (%) | 98.6 (62.0) |
| <I>/<σ(I)> | 8.2 (2.2) |
| Completeness (%) | 99.8 (99.8) |
| Redundancy | 4.8 (4.9) |
| Total observations | 111229 (5581) |
| Unique reflections | 23059 (1129) |
| Refinement | |
| R$_{work}$ (%) | 15.3 |
| R$_{free}$ (%) | 21.4 |
| Solvent content (%) | 38.0 |
| # atoms | |
| Protein | 1399 |
| Ligand/ion | 20 |
| Water | 94 |
| B-factors (Å$^2$) | |
| Protein | 26.4 |
| Ligand/ion | 50.5 |
| Water | 37.7 |
| R.m.s deviations | |
| Bonds (Å) | 0.015 |
| Angles (°) | 1.79 |

*Where values in parentheses refer to the highest resolution shell.

majority of the homodimer interface (*Figure 6A*), and these residues are highlighted in *Figure 6B*. In agreement with the IreB structural analysis, 1200 Å$^2$ of surface area is buried in the ReoM dimer interface, representing 9% of the total solvent accessible surface area. The similarity of the monomers and the dimeric assemblies of ReoM and IreB is underlined by the 1.5 and 1.7 Å r.m.s.d. values, respectively, on global secondary structure superposition matching 74 Cα atoms from each protomer in the comparison.

Other than the compact helical bundle of ReoM, there is a ~ 16 residue-long N-terminal tail, with B-factors 25% higher than the rest of the protein, prior to the start of α-helix one at residue Ile17. The equivalent N-terminal region is also disordered in the NMR structure of IreB (*Hall et al., 2017*). Despite the absence of secondary structure, the ReoM model covering this region could be built with confidence from Asp5 in chain A and Asp2 in chain B (*Figure 6C*). Consequently, it is possible to visualise Thr7, the target for phosphorylation by PrkA, in the flexible N-terminal region in both chains. The side chain of Thr7 in both chains makes no intramolecular interactions and is thus amenable to phosphorylation by PrkA. Despite multiple attempts, however, no crystals of P-ReoM could be grown. Several possible ReoM responses to phosphorylation exist including a change in oligomeric state, as observed quite commonly in response regulators in order to bind more effectively to promoter regions to effect transcription (*Johnson and Lewis, 2001*). However, analysis of the oligomeric state of P-ReoM by size exclusion chromatography revealed that the protein behaved in solution the same as to unphosphorylated ReoM (*Figure 6—figure supplement 1*).

Alternatively, the presence of a sulphate ion (a component of the crystallisation reagent) adjacent to the sidechain of Thr7 could mimic what P-ReoM might look like (*Figure 6C*). The sulphate ion is captured by a positively-charged micro-environment from a symmetry-equivalent molecule. ReoM could thus react to phosphorylation by a substantial movement of Thr7 to interact with this positively-charged surface, which comprises arginines with levels of conservation (Arg57 [57% conserved], Arg62 [99%], Arg66 [76%], Arg70 [98%]) amongst all 2909 ReoM homologues present at NCBI approaching that of Thr7 (96%). We subsequently made alanine substitutions of each of these arginines in *reoM*. Whereas the R66A and R70A mutations were without any effect on growth (data not shown), expression of ReoM R57A and R62A mutations were as lethal as expression of ReoM T7A (*Figure 6—figure supplement 2*). Thus, Arg57 and Arg62 might co-ordinate P-Thr7, stabilising the conformation and position of the flexible N-terminal region (*Figure 6—figure supplement 3*), though confirmation of the molecular consequences of ReoM phosphorylation remain to be determined.

## Control of MurA stability and PG biosynthesis by the PrkA/PrpC protein kinase/phosphatase pair

To study the contribution of the PrkA/PrpC couple to PG biosynthesis in more detail, we aimed to construct *prkA* and *prpC* deletion mutants, but failed to delete *prkA*. However, *prkA* could be deleted in the presence of an IPTG-inducible ectopic *prkA* copy and the resulting strain (LMSW84) required IPTG for growth (*Figure 7A*), demonstrating the essentiality of this gene. The essentiality of *prkA* in our hands is consistent with results by others who have also shown that *prkA* can only be inactivated in the presence of a second copy (*Pensinger et al., 2014*). Repeated attempts to delete *prpC* finally yielded a single Δ*prpC* clone (LMSW76). Genomic sequencing of this strain, which grew at a similar rate to wild type (*Figure 7A*), confirmed the successful deletion of *prpC* but also identified a trinucleotide deletion in the *prkA* gene (designated *prkA\**), effectively removing the complete codon of Gly18 that is part of a conserved glycine-rich loop important for ATP binding (*Rakette et al., 2012*). Presumably, this mutation reduces the PrkA kinase activity to balance the absence of PrpC. By contrast, *prpC* could be deleted readily in the presence of a second IPTG-dependent copy of *prpC* and growth of the resulting strain (LMSW83) did not require IPTG, most likely explained by promoter leakiness in the absence of IPTG (*Figure 7A*). The viability of the i*prpC* mutant shows that the *prpC* deletion had no polar effects on the expression of the downstream *prkA*. That *prkA* and *prpC* are both essential suggests that some of their substrates must be phosphorylated and unphosphorylated, respectively, to be active. Next, the effect of *prkA* and *prpC* mutations on MurA accumulation was analysed by western blotting. Intermediate MurA accumulation was evident in the Δ*prpC prkA\** strain, while full accumulation of MurA was observed in PrpC-depleted cells. By contrast, no MurA was detected in cells depleted for PrkA (*Figure 7B*). Therefore, PrkA and PrpC inversely contribute to the accumulation of MurA, suggesting that phosphorylated

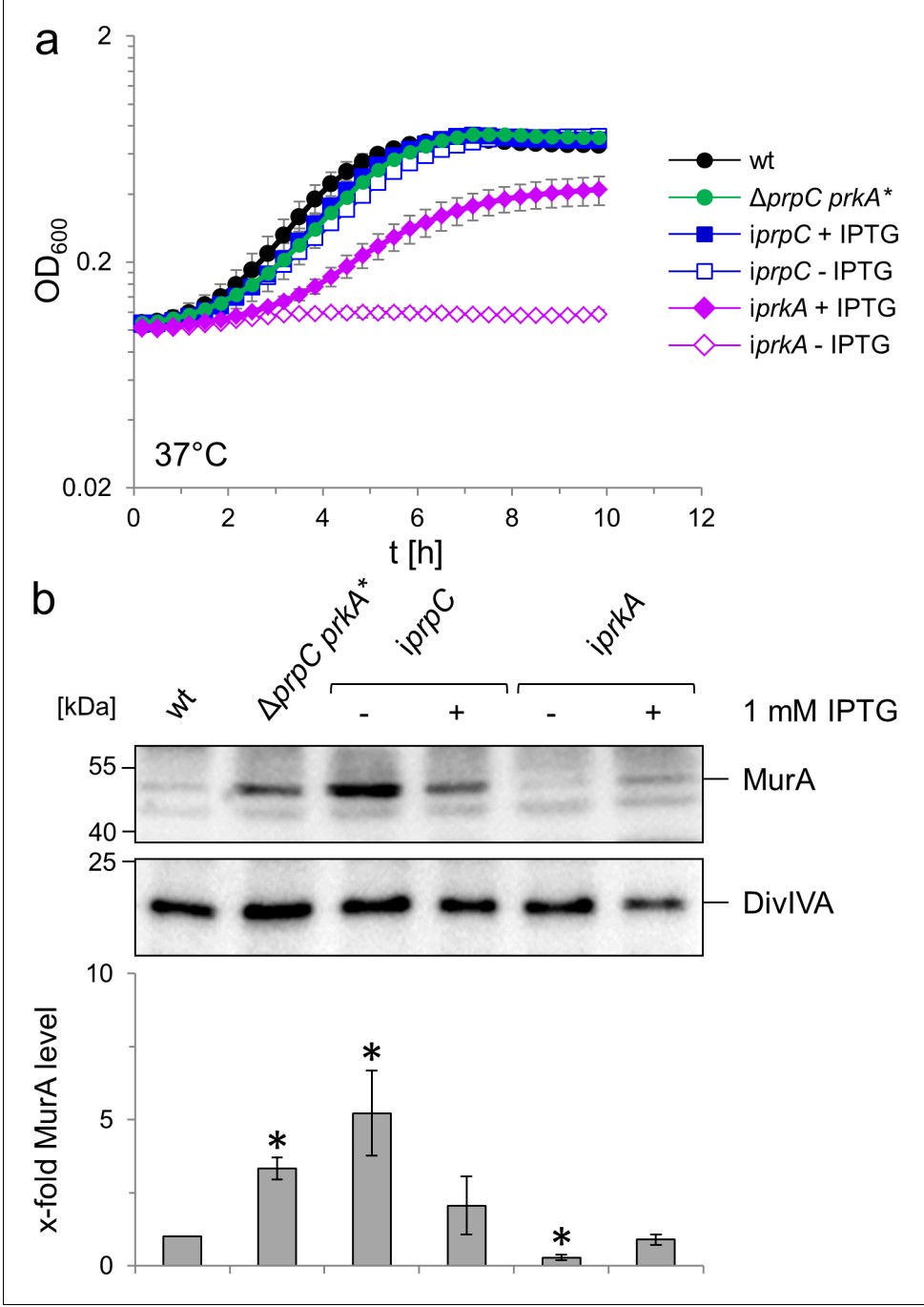

**Figure 7.** Effect of *prkA* and *prpC* mutations on growth and MurA levels of *L. monocytogenes*. (**A**) Contribution of PrkA and PrpC to *L. monocytogenes* growth. *L. monocytogenes* strains EGD-e (wt), LMSW76 (Δ*prpC prkA\**), LMSW83 (i*prpC*) and LMSW84 (i*prkA*) were grown in BHI broth ±1 mM IPTG at 37°C in a microtiter plate reader. The experiment was repeated three times and average values and standard deviations are shown. (**B**) Contribution of PrkA and PrpC to MurA stability. Western blots showing cellular levels of MurA (top) and DivIVA (middle) in the same set of strains as in panel A and quantification of MurA signals by densitometry (below). Average values and standard deviations calculated from three independent experiments are shown. Asterisks indicate statistically significant differences (p<0.05, *t*-test).

ReoM favours MurA accumulation, while un-phosphorylated ReoM counteracts this process. In good agreement, depletion of PrkA strongly increased ceftriaxone susceptibility, while inactivation of *prpC* caused increased ceftriaxone resistance (*Figure 3A*).

## Deletion of *reoM*, *reoY* or *clpC* eliminates *prkA* essentiality

In order to test whether the essentiality of *prkA* could be explained by stimulated MurA degradation through ClpCP, we first tested the effect of *clpC* on the essentiality of *prkA*. For this purpose, *clpC* was removed from the i*prkA* strain and growth of the resulting strain (LMSW91) was tested. In contrast to the parental i*prkA* strain (LMSW84), which required IPTG for growth, strain LMSW91 was viable without IPTG (*Figure 8A*) thus confirming that the essentiality of PrkA depends on ClpC. We next wondered whether ReoM and ReoY were also required for PrkA essentiality and consequently deleted their genes from the i*prkA* background to test this. Again, the resulting strains did not require IPTG for growth in contrast to the parental i*prkA* strain (*Figure 8A*). In good agreement with these findings, deletion of *clpC*, *reoM* or *reoY* all stabilised MurA in PrkA-depleted cells (*Figure 8B*), showing that the stimulated degradation of MurA that we observe in cells depleted for PrkA (*Figure 7B*) is dependent on any one of these three proteins. These results together permit a model of genetic interactions to be proposed (*Figure 9*) that starts with PrkA and its downstream substrate ReoM. ReoY, MurZ and ClpC in turn are positioned downstream of ReoM (as indicated by the experiments shown in *Figure 5E*) to control MurA stability. To further substantiate this concept, physical interactions between ReoM, ReoY, ClpC, ClpP and MurA were determined in bacterial two hybrid experiments, which revealed that ReoY interacted with ClpC, ClpP and ReoM. In turn, ReoM interacted with MurA (*Figure 8—figure supplement 1*), which suggests that ReoM and ReoY could bridge the interaction of ClpCP with its substrate MurA.

## Discussion

With ReoM we have identified a missing link in a regulatory pathway that enables Firmicute bacteria to respond to PG biosynthesis fluctuations associated with growth and division. In *L. monocytogenes*, the sensory module of this pathway comprises the membrane integral protein kinase PrkA and the cognate protein phosphatase PrpC, their newly discovered substrate ReoM and the associated factors ReoY and MurZ, which together regulate ClpCP activity, the effector protease that acts on MurA (*Figure 9*). It has been demonstrated previously that the kinase activity of PrkA homologues was activated by muropeptides (*Mir et al., 2011*; *Shah et al., 2008*) or the PG precursor lipid II (*Hardt et al., 2017*). Muropeptides were released from the cell wall during normal PG turnover, and their release was intensified when PG hydrolysis prevailed over PG biosynthesis (*Vollmer et al., 2008b*; *Irazoki et al., 2019*), whereas blocking PG chain elongation by moenomycin treatment caused the accumulation of lipid-linked PG precursors (*Kohlrausch and Höltje, 1991*). Thus, both types of molecules accumulated when PG biosynthesis was inhibited and could represent useful signals for detecting imbalances in cell wall biosynthesis. Our data are consistent with a model in which PrkA-phosphorylated ReoM no longer activates ClpCP, which leads to MurA stabilisation and the activation of PG biosynthesis (*Figure 9*). In *B. subtilis*, this effect is supported by stabilisation of GlmS (*Figure 2—figure supplement 3A*), another ClpCP substrate but which acts in front of MurA as the first enzyme of the UDP-Glc*N*Ac-generating GlmSMU pathway.

How ReoM and ReoY exert their effect on ClpCP is currently unknown, but a fascinating possibility would be a function like to that of an adaptor protein to target protein substrates to ClpCP for degradation. Several ClpC adaptors for different substrates are known in *B. subtilis* (*Kirstein et al., 2009*; *Mulvenna et al., 2019*), but an adaptor for *Bs*MurAA is not among them (*Kock et al., 2004*; *Kirstein et al., 2009*). Like ReoM, the ClpC adaptor McsB from *B. subtilis* is also subject to phosphorylation, but - unlike ReoM - it targets its substrate CtsR to the ClpCP machinery only when phosphorylated (*Kirstein et al., 2007*). Either ReoM or ReoY could act as this adaptor, leaving a subsidiary function for the other respective protein. Alternatively, both proteins could work in tandem, where each of them is equally needed for ClpCP-dependent MurA degradation since the phenotypes of *reoM* and *reoY* mutants were identical with respect to MurA stability. However, overexpression or deletion of *reoM* altered the phenotype of the Δ*gpsB* mutant, but that of *reoY* was without phenotype (*Figure 1—figure supplement 1*, *Figure 1—figure supplement 2*). ReoY, restricted to the *Bacilli*, also showed a narrower phylogenetic distribution than ReoM, which is found

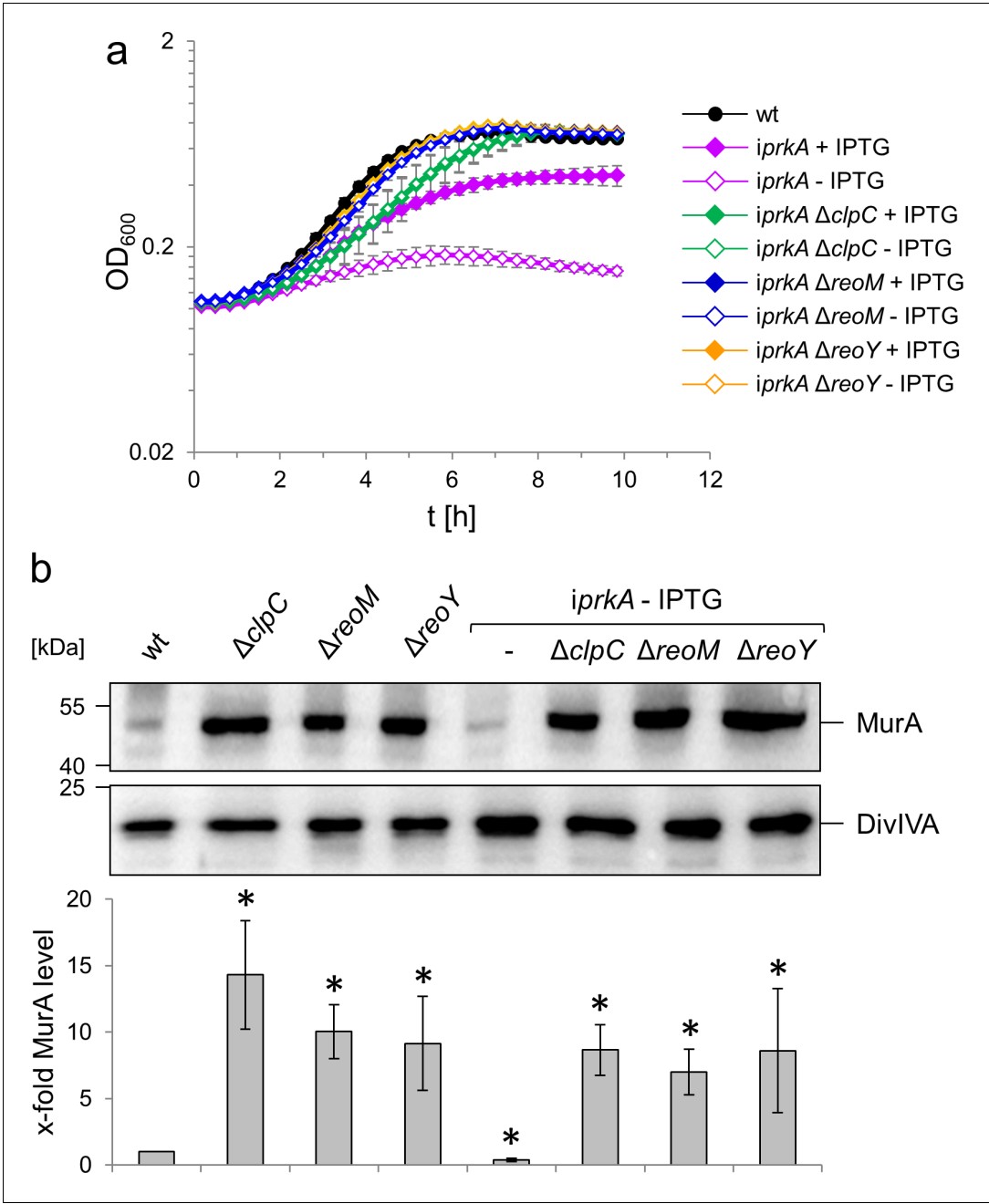

**Figure 8.** PrkA essentiality depends on *reoM*, *reoY* and *clpC*. (**A**) Effect of *reoM*, *reoY* and *clpC* deletions on *prkA* essentiality. *L. monocytogenes* strains EGD-e (wt), LMSW84 (i*prkA*), LMSW89 (i*prkA* Δ*reoM*), LMSW90 (i*prkA* Δ*reoY*) and LMSW91 (i*prkA* Δ*clpC*) were grown in BHI broth ±1 mM IPTG at 37°C in a microtiter plate reader. The experiment was repeated three times and average values and standard deviations are shown. (**B**) *clpC, reoM* and *reoY* deletions overcome MurA degradation in PrkA-depleted cells. Western blot showing MurA levels in *L. monocytogenes* strains EGD-e (wt), LMJR138 (Δ*clpC*), LMSW30 (Δ*reoM*), LMSW32 (Δ*reoY*), LMSW84 (i*prkA*), LMSW89 (i*prkA* Δ*reoM*), LMSW90 (i*prkA* Δ*reoY*) and LMSW91 (i*prkA* Δ*clpC*, top). A parallel DivIVA western blot was used as loading control (middle). Quantification of MurA signals by densitometry (below). Average values and standard deviations calculated from three independent experiments are shown. Asterisks indicate statistically significant differences (p<0.05, *t*-test).

The online version of this article includes the following figure supplement(s) for figure 8:

**Figure supplement 1.** Bacterial two hybrid experiment showing interactions between MurA, ReoM, ReoY, ClpC and ClpP.

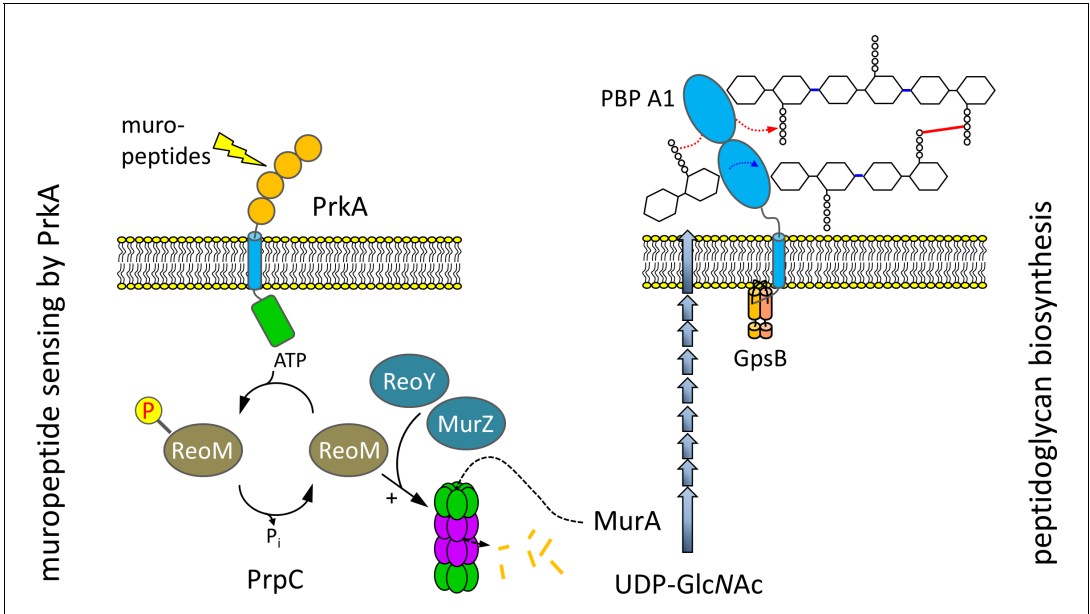

**Figure 9.** ReoM links PrkA-dependent muropeptide sensing with peptidoglycan biosynthesis. Model illustrating the role of ReoM as substrate of PrkA and as regulator of ClpCP. PrkA recognises free muropeptides, which activate PrkA to phosphorylate ReoM. In its unphosphorlyated form, ReoM is an activator of ClpCP-dependent degradation of MurA, the first enzyme of peptidoglycan biosynthesis, and ReoY and MurZ contribute to this process. By phosphorylating ReoM, PrkA prevents ClpCP-dependent MurA degradation so that MurA accumulates and peptidoglycan biosynthesis can occur. Please note that there is a lesser degree of conservation in the fourth PASTA domain of PrkA.

across different *Firmicutes* (*Figure 6B*). Thus, it seems that ReoM might have a more generalised role, whereas ReoY could play a subordinate function in control of MurA degradation by ClpCP. The role of the MurA homologue MurZ in this process is entirely unclear, but our genetic data now place it downstream of ReoM (*Figure 8C*). Furthermore, arginine phosphorylation targets proteins to ClpCP for degradation (*Trentini et al., 2016*). *L. monocytogenes* MurA contains 17 arginines and MurAA of *B. subtilis* has been found in complex with the protein arginine phosphatase YwlE (*Elsholz et al., 2012*). The possibility that MurA proteins could also require arginine phosphorylation to be accepted as a substrate by ClpCP offers additional control possibilities for ReoM/ReoY/MurZ to modulate MurA levels.

*L. monocytogenes prkA* is essential, but *prkA* homologues in other species are dispensable (*Gaidenko et al., 2002*; *Cuenot et al., 2019*; *Kristich et al., 2007*; *Débarbouillé et al., 2009*; *Nováková et al., 2005*). At least in some of them (such as *E. faecalis*, *S. aureus* and *S. pneumoniae*), the primary MurA enzyme can be functionally replaced by a second paralogue (*Blake et al., 2009*; *Vesić and Kristich, 2012*; *Du et al., 2000*), so that proteolytic degradation of the primary enzyme can be tolerated. By contrast, *prkC* is dispensable to *Clostridioides difficile* despite encoding only one copy of the essential MurA gene (*Cuenot et al., 2019*; *Sapkota et al., 2020*); *B. subtilis prkC* can also be deleted even though the primary MurA enzyme cannot be replaced by the secondary one (*Kock et al., 2004*; *Gaidenko et al., 2002*). Probably, control of MurA degradation by ClpCP is somewhat relaxed in these latter two species.

A screen for *gpsB* suppressors in *S. pneumoniae* did not yield *reoM* mutations (and these strains do not contain *reoY*, consistent with a subordinate function for this gene), but instead suppressor mutations were found that affect *phpP*, which encodes a Ser/Thr protein phosphatase that acts in concert with StkP, the PASTA-eSTK of this organism (*Rued et al., 2017*; *Lewis, 2017*). Absence or inactivation of PhpP triggered an increase in StkP-dependent protein phosphorylation levels in the pneumococcus (*Rued et al., 2017*; *Ulrych et al., 2016*). It is tempting to speculate that loss of PhpP activity in this suppressor also triggers P-ReoM formation that, according to our model, would help to stabilise MurA and thus suppress the ΔgpsB phenotype. Interestingly, another *S. pneumoniae gpsB* suppressor was identified that carries a duplication of a ~ 150 kb genomic fragment (*Rued et al., 2017*), a region that includes the open reading frame for MurA. Suppression of the

*gpsB* phenotype in this instance could also work via MurA accumulation, but this time due to a gene dosage effect.

It is becoming increasingly evident that control of PG biosynthesis in response to cell wall derived signals, via PASTA-eSTKs, is a regulatory capacity common to *Firmicutes* and *Actinobacteria*. CwlM is the critical kinase substrate in the actinobacterium *M. tuberculosis* that, when phosphorylated by PknB, binds to and activates MurA (*Boutte et al., 2016*). Homologues of CwlM are not present in *L. monocytogenes* or *B. subtilis* and instead these bacteria adjust their MurA levels by controlling MurA turnover in response to PrkA-dependent phosphorylation of ReoM. Consequently, both mechanisms activate PG biosynthesis in a PrkA-dependent manner either by activation or stabilisation of MurA. Presumably *B. subtilis*, and other endospore forming bacteria, re-start PG biosynthesis at the onset of germination in a similar way. Germination of *B. subtilis* spores can be triggered by muropeptides in a manner that depends upon PrkC (*Shah et al., 2008*), the PASTA-eSTK of *B. subtilis* (*Madec et al., 2002*). Even though *Bs*PrkC phosphorylates multiple substrates (*Ravikumar et al., 2014*), whose individual contribution to germination is not known precisely, phosphorylation of ReoM (*aka* YrzL) could be required to restart PG biosynthesis in germinating *B. subtilis* cells by stabilising MurAA. Moreover, an *E. faecalis* mutant lacking the PASTA-eSTK IreK was more susceptible to ceftriaxone but overexpression of *Ef*MurAA overcame this defect (*Vesić and Kristich, 2012*). This implies the possibility that unphosphorylated IreB together with the ReoY homologue of this organism, OG1RF_11272 (*Banla et al., 2018*), might stimulate MurAA proteolysis in *E. faecalis* as well. Taken together it seems that observations made in different *Firmicutes* are in good agreement with the PrkA→ReoM/ReoY→ClpC→MurA signaling sequence that we propose. The open questions that remain on the molecular mechanism of ClpCP control by ReoM and ReoY will be addressed by future experiments.

## Materials and methods

### Bacterial strains and growth conditions

*Table 2* lists all strains used in this study (also see *Supplementary file 1*). Strains of *L. monocytogenes* were cultivated in BHI broth or on BHI agar plates. *B. subtilis* strains were grown in LB broth at 37°C. Antibiotics and supplements were added when required at the following concentrations: erythromycin (5 µg/ml), kanamycin (50 µg/ml), X-Gal (100 µg/ml) and IPTG (as indicated). *Escherichia coli* TOP10 was used as host for all cloning procedures (*Sambrook et al., 1989*). Minimal inhibitory concentrations against ceftriaxone were determined as described previously (*Rismondo et al., 2015*) using E-test strips with a ceftriaxone concentration range of 0.016–256 µg/ml.

### General methods, manipulation of DNA and oligonucleotide primers

All key resources used in this study are listed in *Supplementary file 1*. Standard methods were used for transformation of *E. coli* and isolation of plasmid DNA (*Sambrook et al., 1989*). Transformation of *L. monocytogenes* was carried out as described by others (*Monk et al., 2008*). Restriction and ligation of DNA was performed according to the manufacturer´s instructions. All primer sequences are listed in *Table 3* (also see *Supplementary file 1*).

### Construction of plasmids for recombinant protein expression

The plasmids for expressing recombinant versions of ReoM, PrkA-KD and PrpC were prepared by first amplifying the corresponding genes (*reoM*, *lmo1820* and *lmo1821*) from *L. monocytogenes* EGD-e genomic DNA using primer pairs Lmo1503F/Lmo1503R, PrkAF/PrkAR, and PrpCF/PrpCR, respectively. The PCR products were individually ligated between the NcoI and XhoI sites of pETM11 (*Peränen et al., 1996*). All mutagenesis was carried out using the Quikchange protocol and the correct sequence of each plasmid and mutant constructed was verified by Sanger DNA sequencing (Eurofins Genomics).

### Construction of plasmids for generation of *L. monocytogenes* strains

Plasmid pJR65 was constructed for the inducible expression of *reoM*. To this end, the *reoM* open reading frame was amplified by PCR using the oligonucleotides JR169/JR170 and cloned into pIMK3 using NcoI/SalI. The T7A and T7D mutations were introduced into *reoM* of plasmid pJR65 by

**Table 2.** Plasmids and strains used in this study.

| Name | Relevant characteristics | Source*/reference |
|---|---|---|
| Plasmids | | |
| pIMK3 | $P_{help}$-lacO lacI neo | *Monk et al., 2008* |
| pMAD | bla erm bgaB | *Arnaud et al., 2004* |
| pUT18 | bla $P_{lac}$-cya(T18) | *Karimova et al., 1998* |
| pUT18C | bla $P_{lac}$-cya(T18) | *Karimova et al., 1998* |
| pKT25 | kan $P_{lac}$-cya(T25) | *Karimova et al., 1998* |
| p25-N | kan $P_{lac}$-cya(T25) | *Claessen et al., 2008* |
| pJR127 | bla erm bgaB ΔclpC (lmo0232) | *Rismondo et al., 2017* |
| pSH246 | bla erm bgaB ΔgpsB (lmo1888) | *Rismondo et al., 2016* |
| pJR68 | bla erm bgaB ΔmurZ (lmo2552) | *Rismondo et al., 2017* |
| pJR71 | $P_{help}$-lacO-murZ lacI neo | *Rismondo et al., 2017* |
| pJR65 | $P_{help}$-lacO-reoM lacI neo | this work |
| pJR70 | $P_{help}$-lacO-reoY lacI neo | this work |
| pJR83 | bla erm bgaB ΔreoY (lmo1921) | this work |
| pJR101 | kan $P_{lac}$-cya(T25)-reoM | this work |
| pJR102 | kan $P_{lac}$-reoM-cya(T25) | this work |
| pJR103 | bla $P_{lac}$-reoM-cya(T18) | this work |
| pJR104 | bla $P_{lac}$-cya(T18)-reoM | this work |
| pJR109 | kan $P_{lac}$-cya(T25)-reoY | this work |
| pJR111 | bla $P_{lac}$-cya(T18)-reoY | this work |
| pJR116 | kan $P_{lac}$-cya(T25)-murA | this work |
| pJR117 | kan $P_{lac}$-murA-cya(T25) | this work |
| pJR118 | bla $P_{lac}$-murA-cya(T18) | this work |
| pJR119 | bla $P_{lac}$-cya(T18)-murA | this work |
| pJR121 | bla $P_{lac}$-reoY-cya(T18) | this work |
| pJR126 | bla erm bgaB ΔreoM (lmo1503) | this work |
| pSW29 | $P_{help}$-lacO-reoM T7A lacI neo | this work |
| pSW30 | $P_{help}$-lacO-reoM T7D lacI neo | this work |
| pSW36 | bla erm bgaB ΔprkA (lmo1820) | this work |
| pSW37 | bla erm bgaB ΔprpC (lmo1821) | this work |
| pSW38 | $P_{help}$-lacO-prkA lacI neo | this work |
| pSW39 | $P_{help}$-lacO-prpC lacI neo | this work |
| pSW43 | kan $P_{lac}$-cya(T25)-clpC | this work |
| pSW44 | kan $P_{lac}$-cya(T25)-clpP | this work |
| pSW45 | kan $P_{lac}$- clpC-cya(T25) | this work |
| pSW46 | kan $P_{lac}$-clpP-cya(T25) | this work |
| pSW47 | bla $P_{lac}$-clpC-cya(T18) | this work |
| pSW48 | bla $P_{lac}$-clpP-cya(T18) | this work |
| pSW49 | bla $P_{lac}$-cya(T18)-clpC | this work |
| pSW50 | bla $P_{lac}$-cya(T18)-clpP | this work |
| pSW55 | $P_{help}$-lacO-reoM R66A lacI neo | this work |
| pSW56 | $P_{help}$-lacO-reoM R70A lacI neo | this work |
| pSW58 | $P_{help}$-lacO-reoM R57A lacI neo | this work |
| pSW59 | $P_{help}$-lacO-reoM R62A lacI neo | this work |

*Table 2 continued on next page*

*Table 2 continued*

| Name | Relevant characteristics | Source*/reference |
|---|---|---|
| *B. subtilis* strains | | |
| 168 | wild type, lab collection | |
| BKE00860 | ΔclpC | *Koo et al., 2017* |
| BKE22180 | ΔgpsB | *Koo et al., 2017* |
| BKE22580 | ΔypiB (reoY) | *Koo et al., 2017* |
| BKE27400 | ΔyrzL (reoM) | *Koo et al., 2017* |
| *L. monocytogenes* strains | | |
| EGD-e | wild-type, serovar 1/2a strain | *Glaser et al., 2001* |
| LMJR19 | ΔgpsB (lmo1888) | *Rismondo et al., 2016* |
| LMJR104 | ΔmurZ (lmo2552) | *Rismondo et al., 2017* |
| LMJR116 | attB::$P_{help}$-lacO-murA lacI neo | *Rismondo et al., 2017* |
| LMJR123 | ΔmurA (lmo2526) attB::$P_{help}$-lacO-murA lacI neo | *Rismondo et al., 2017* |
| LMJR138 | ΔclpC (lmo0232) | *Rismondo et al., 2017* |
| *shg8* | ΔgpsB reoY H87Y | this work |
| *shg10* | ΔgpsB reoY TAA74 | this work |
| *shg12* | ΔgpsB reoM RBS mutation | this work |
| LMJR96 | ΔgpsB attB::$P_{help}$-lacO-reoM lacI neo | pJR65 → LMJR19 |
| LMJR102 | attB::$P_{help}$-lacO-reoM lacI neo | pJR65 → EGD-e |
| LMJR106 | ΔgpsB attB::$P_{help}$-lacO-reoY lacI neo | pJR70 → LMJR19 |
| LMJR120 | ΔgpsB ΔreoY | pJR83 ↔ LMJR19 |
| LMJR137 | ΔgpsB ΔreoM | pJR126 ↔ LMJR19 |
| LMJR171 | ΔclpC ΔmurZ | pJR127 ↔ LMJR104 |
| LMSW30 | ΔreoM (lmo1503) | pJR126 ↔ EGD-e |
| LMSW32 | ΔreoY (lmo1921) | pJR83 ↔ EGD-e |
| LMSW50 | ΔclpC ΔreoM | pJR127 ↔ LMSW30 |
| LMSW51 | ΔclpC ΔreoY | pJR127 ↔ LMSW32 |
| LMSW52 | ΔreoM attB::$P_{help}$-lacO-reoM T7A lacI neo | pSW29 → LMSW30 |
| LMSW53 | ΔreoM attB::$P_{help}$-lacO-reoM T7D lacI neo | pSW30 → LMSW30 |
| LMSW57 | ΔreoM attB::$P_{help}$-lacO-reoM lacI neo | pJR65 → LMSW30 |
| LMSW72 | ΔreoM attB::$P_{help}$-lacO-reoM T7A lacI neo ΔclpC | pJR127 ↔ LMSW52 |
| LMSW76 | ΔprpC prkA[*] | pSW37 ↔ EGD-e |
| LMSW80 | attB::$P_{help}$-lacO-prkA lacI neo | pSW38 → EGD-e |
| LMSW81 | attB::$P_{help}$-lacO-prpC lacI neo | pSW39 → EGD-e |
| LMSW83 | ΔprpC attB::$P_{help}$-lacO-prpC lacI neo | pSW37 ↔ LMSW81 |
| LMSW84 | ΔprkA attB::$P_{help}$-lacO-prkA lacI neo | pSW36 ↔ LMSW80 |
| LMSW89 | ΔprkA attB::$P_{help}$-lacO-prkA lacI neo ΔreoM | pJR126 ↔ LMSW84 |
| LMSW90 | ΔprkA attB::$P_{help}$-lacO-prkA lacI neo ΔreoY | pJR83 ↔ LMSW84 |
| LMSW91 | ΔprkA attB::$P_{help}$-lacO-prkA lacI neo ΔclpC | pJR127 ↔ LMSW84 |
| LMSW117 | ΔreoM ΔreoY | pJR126 ↔ LMSW32 |
| LMSW118 | ΔreoY ΔmurZ | pJR68 ↔ LMSW32 |
| LMSW119 | ΔreoM ΔmurZ | pJR68 ↔ LMSW30 |
| LMSW120 | ΔreoM attB::$P_{help}$-lacO-reoM R66A lacI neo | pSW55 → LMSW30 |
| LMSW121 | ΔreoM attB::$P_{help}$-lacO-reoM R70A lacI neo | pSW56 → LMSW30 |
| LMSW123 | ΔreoM attB::$P_{help}$-lacO-reoM T7A lacI neo ΔreoY | pSW29 → LMSW117 |

*Table 2 continued on next page*

*Table 2 continued*

| Name | Relevant characteristics | Source*/reference |
|------|--------------------------|-------------------|
| LMSW124 | ΔreoM attB::P$_{help}$-lacO-reoM T7A lacI neo ΔmurZ | pSW29 → LMSW119 |
| LMSW125 | ΔreoM attB::P$_{help}$-lacO-reoM R57A lacI neo | pSW58 → LMSW30 |
| LMSW126 | ΔreoM attB::P$_{help}$-lacO-reoM R62A lacI neo | pSW59 → LMSW30 |
| LMSW138 | ΔreoY attB::P$_{help}$-lacO-reoY lacI neo | pJR70 → LMSW32 |
| LMSW139 | ΔmurZ attB::P$_{help}$-lacO-murZ lacI neo | pJR71 → LMJR104 |

*The arrow (→) stands for a transformation event and the double arrow (↔) indicates gene deletions obtained by chromosomal insertion and subsequent excision of pMAD plasmid derivatives (see experimental procedures for details).

quickchange mutagenesis using the primer pair SW77/SW78 and SW79/SW80, respectively. The R57A, R62A R66A and R70A, mutations were introduced into pJR65 in the same way, but using primer pairs SW144/SW145, SW146/SW147, SW136/SW137 and SW138/SW139, respectively.

Plasmid pJR70 was constructed for inducible *reoY* expression. For this purpose, *reoY* was amplified using the primer pair JR163/JR164 and cloned into pIMK3 using NcoI/SalI.

Plasmid pSW38, for IPTG-inducible *prkA* expression, was constructed by amplification of *prkA* using the oligonucleotides SW112/SW113 and the subsequent cloning of the generated fragment into pIMK3 using BamHI/SalI. Plasmid pSW39, for IPTG-controlled expression of *prpC*, was constructed analogously, but using the oligonucleotides SW110/SW111 as the primers.

For construction of plasmid pJR83, facilitating deletion of *reoY*, fragments encompassing ~800 bp up- and down-stream of *reoY* were amplified by PCR with the primer pairs JR197/JR198 and JR199/JR200. Both fragments were spliced together by splicing by overlapping extension (SOE) PCR and cloned into pMAD using BamHI/EcoRI.

Plasmid pJR126 was generated for deletion of *reoM*. Fragments up- and down-stream of *reoM* were PCR amplified using the primers JR264/JR265 and JR266/JR267, respectively. Both fragments were cut with BamHI, fused together by ligation and the desired fragment was amplified from the ligation mixture by PCR using the primers JR264/JR267 and then cloned into pMAD using BglII/SalI.

Plasmid pSW36 was constructed to delete the *prkA* gene. Fragments up- and down-stream of *prkA* were amplified in separate PCRs using the primer pairs SHW819/SHW821 and SHW820/SHW822, respectively. Both fragments were fused together by SOE-PCR and inserted into pMAD by restriction free cloning (*van den Ent and Löwe, 2006*). Plasmid pSW37, facilitating deletion of *prpC*, was constructed in a similar manner. Up- and down-stream fragments of *prpC* were amplified using the primer pairs SHW815/SHW817 and SHW816/SHW818 and fused together by SOE-PCR. The resulting fragment was inserted into pMAD by restriction free cloning.

Derivatives of pIMK3 were introduced into *L. monocytogenes* strains by electroporation and clones were selected on BHI agar plates containing kanamycin. Plasmid insertion at the *attB* site of the tRNA$^{Arg}$ locus was verified by PCR. Plasmid derivatives of pMAD were transformed into the respective *L. monocytogenes* recipient strains and genes were deleted as described elsewhere (*Arnaud et al., 2004*). All gene deletions were confirmed by PCR.

## Construction of bacterial two hybrid plasmids

The *reoM* (JR255/JR256), *reoY* (JR253/JR254), *clpC* (SHW830/831) and *clpP* (SHW832/833) genes were amplified using the primer pairs given in brackets and cloned into pUT18, pUT18C, pKT25 and p25-N plasmids using XbaI/KpnI. The *murA* gene was amplified using the oligonucleotides JR249/JR250 for cloning into pKT25 and p25-N using PstI/KpnI or using the JR257/JR250 primer pair for cloning into pUT18 and pUT18C using the same restriction enzymes.

## Bacterial two hybrid experiments

Plasmids carrying genes fused to T18- or the T25-fragments of the *Bordetella pertussis* adenylate cyclase were co-transformed into *E. coli* BTH101 (*Karimova et al., 1998*) and transformants were selected on LB agar plates containing ampicillin (100 μg ml$^{-1}$), kanamycin (50 μg ml$^{-1}$), X-Gal (0.004%) and IPTG (0.1 mM). Agar plates were photographed after 48 hr of incubation at 30°C.

**Table 3.** Oligonucleotides used in this study.

| Name | Sequence (5´→3´) |
| --- | --- |
| JR163 | GCGCCCATGGCTAAGGCATCCATTTCAATAGACGAGAAG |
| JR164 | GCGCGTCGACTTATTCTTTTTCCGTATCCATTTGCTGTA |
| JR169 | GCGCCCATGGATTCAAAAGATCAAACAATGTTTTACAACTTC |
| JR170 | GCGCGTCGACTCATTTCTCACCAATTTCGTTATTTTTCAG |
| JR197 | GCGCGGATCCCAATTATTTCGAATGGTGCGGTGTC |
| JR198 | TCCTTATTCGTCGACCATCTTTCCTCAGTCCCTTCCTG |
| JR199 | GGAAAGATGGTCGACGAATAAGGAATAAATCCTAGTTAGTAGGG |
| JR200 | CGCGCGAATTCCCAAGACTCAACCTCTTTCACTC |
| JR249 | GCGCCTGCAGAAAAAATTATTGTACGCGGTGGAAAAC |
| JR250 | GCGCGGTACCGCGAATAAAGACGCTAAGTTTGTTACATCG |
| JR253 | GCGCTCTAGAAAAGGCATCCATTTCAATAGACGAG |
| JR254 | GCGCGGTACCTCTTTTTCCGTATCCATTTGCTG |
| JR255 | GCGCTCTAGATTCAAAAGATCAAACAATGTTTTACAAC |
| JR256 | GCGCGGTACCTTCTCACCAATTTCGTTATTTTTCAG |
| JR257 | GCGCCTGCAGGGAAAAAATTATTGTACGCGGTGGAAAAC |
| JR264 | GCGCAGATCTGGCAAATACAGCATTGAACTATGTG |
| JR265 | GCGCGGATCCAATCGAAGCACCTCATTCCTTC |
| JR266 | GCGCGGATCCATGAGAATAATGGGTTTAGATGTCGGC |
| JR267 | GCGCGTCGACGCTAGGAATGTAGCAAGGATTTCTTC |
| SHW815 | GATCTATCGATGCATGCCATGGGCTAAATGACCAAGGAATTACCG |
| SHW816 | CGCGTCGGGCGATATCGGATCCTTTCTTCCGCGTTTTGGTAACG |
| SHW817 | CAATCATCATTTTAAAAGCACCTCACTATTTTTCAG |
| SHW818 | TGCTTTTAAAATGATGATTGGTAAGCGATTAAGC |
| SHW819 | GATCTATCGATGCATGCCATGGAGATAGAGGCAGAATAAGACATC |
| SHW820 | CGCGTCGGGCGATATCGGATCCGGTATTTACAACCACTACGTCG |
| SHW821 | CGTTCTTATTTCATGAAGCATCCCTCCCTTTC |
| SHW822 | TGCTTCATGAAATAAGAACGGAGGAAATGTGCTG |
| SHW830 | GCGCGCTCTAGATGGACGATTTACGCAAAGAGCTCAG |
| SHW831 | GCGCGCGGTACCTTAGCTTTTACTTTTTTAGAGGTTGTTTTC |
| SHW832 | GCGCGCTCTAGAAATTCCAACAGTAATTGAACAAACTAGC |
| SHW833 | GCGCGCGGTACCCCTTTTAAGCCAGATTTATTAATGATAATATC |
| SW77 | GTAAAACATTGCTTGATCTTTTGAATCCATGGGTTTCAC |
| SW78 | GATCAAGCAATGTTTTACAACTTCGGCGATGATTC |
| SW79 | GTAAAACATGTCTTGATCTTTTGAATCCATGGGTTTCAC |
| SW80 | GATCAAGACATGTTTTACAACTTCGGCG ATGATTC |
| SW110 | GCGCGCGGATCCATGCATGCAGAATTTAGAACAGATAG |
| SW111 | GCGCGCGTCGACTCATGAAGCATCCCTCCCTTTC |
| SW112 | GCGCGCGGATCCATGATGATTGGTAAGCGATTAAGCG |
| SW113 | GCGCGCGTCGACTTAATTTGGATAAGGGACTGTACCTTC |
| SW136 | CTAAACGAGCTATCATACTTCTAGCATCCTTGTGAC |
| SW137 | GTATGATAGCTCGTTTAGAACGAGATGAAATTATCGAG |
| SW138 | AATTTCATCTGCTTCTAAACGACGTATCATACTTCTAGC |
| SW139 | GTTTAGAAGCAGATGAAATTATCGAGGAACTTGTCAAAG |
| SW144 | CCTTGTGAGCAGGAATATAAGCAGGATCGCCTG |

*Table 3 continued on next page*

*Table 3 continued*

| Name | Sequence (5´→3´) |
|------|------------------|
| SW145 | TATATTCCTGCTCACAAGGATGCTAGAAGTATGATAC |
| SW146 | GTATCATACTTGCAGCATCCTTGTGACGAGGAATATAAG |
| SW147 | GGATGCTGCAAGTATGATACGTCGTTTAGAACGAG |
| Lmo1503F | GCTATACCATGGATTCAAAAGATCAAACAATGTTTTACAAC |
| Lmo1503R | CGATATCTCGAGTCATTTCTCACCAATTTCGTTATTTTTCAG |
| PrkAF | GCTATACCATGGCAATGATGATTGGTAAGCGATTAAGCG |
| PrkAR | CGATATCTCGAGTCATTTTTTCTTTTTCTTATCTTTTTTCTCCTCAGG |
| PrpCF | GCTATACCATGGCAATGCATGCAGAATTTAGAACAGATAGAG |
| PrpCR | CGATATCTCGAGTCATGAAGCATCCCTCCCTTTC |

## Genome sequencing

A total of 1 ng of genomic DNA was used for library generation by the Nextera XT DNA Library Prep Kit according to the manufacturer's recommendations (Illumina). Sequencing was carried out on a MiSeq benchtop sequencer and performed in paired-end modes (2 × 300 bp) using a MiSeq Reagent Kit v3 cartridge (600-cycle kit). Sequencing reads were mapped to the reference genome *L. monocytogenes* EGD-e (NC_003210.1) (*Glaser et al., 2001*) by utilising the Geneious software (Biomatters Ltd.). Variants, representing putative suppressor mutations, were identified using the Geneious SNP finder tool. Genome sequences of *shg8*, *shg10*, *shg12* and LMSW76 were deposited at ENA under study number PRJEB35110 and sample accession numbers ERS3927571 (SAMEA6127277), ERS3927572 (SAMEA6127278), ERS3927573 (SAMEA6127279), and ERS3967687 (SAMEA6167687) respectively.

## Isolation of cellular proteins and western blotting

20 ml cells were harvested by centrifugation, washed with ZAP buffer (10 mM Tris.HCl pH7.5, 200 mM NaCl), resuspended in 1 ml ZAP buffer also containing 1 mM PMSF and disrupted by sonication. Centrifugation was used to remove cellular debris and the supernatant was used as total cellular protein extract. Sample aliquots were separated by standard SDS polyacrylamide gel electrophoresis. Gels were transferred onto positively charged polyvinylidene fluoride membranes by semi-dry transfer. ClpC, DivIVA, GlmS, IlvB and MurA were immune-stained using a polyclonal rabbit antiserum recognising *B. subtilis* ClpC (*Gerth et al., 2004*), DivIVA (*Marston et al., 1998*), GlmS, IlvB (*Gerth et al., 2008*) and MurAA (*Kock et al., 2004*) as the primary antibody and an anti-rabbit immunoglobulin G conjugated to horseradish peroxidase as the secondary one. The ECL chemiluminescence detection system (Thermo Scientific) was used for detection of the peroxidase conjugates on the PVDF membrane in a chemiluminescence imager (Vilber Lourmat). For depletion of PrkA, PrkA depletion strains were grown overnight in the presence of 1 mM IPTG and then again inoculated in BHI broth containing 1 mM IPTG to an $OD_{600}$ = 0.05x00A0 and grown for 3 hr at 37°C. Subsequently, cells were centrifuged, washed and reinoculated in BHI broth without IPTG at the same $OD_{600}$ as before centrifugation. Finally, cells were harvested after 3.5 more hours of growth at 37°C and cellular proteins were isolated.

## Microscopy

Cytoplasmic membranes of exponentially growing bacteria were stained through addition of 1 μl of nile red solution (100 μg ml$^{-1}$ in DMSO) to 100 μl of culture. Images were taken with a Nikon Eclipse Ti microscope coupled to a Nikon DS-MBWc CCD camera and processed using the NIS elements AR software package (Nikon) or ImageJ. Ultrathin section transmission electron microscopy and scanning electron microscopy were performed essentially as described earlier (*Rismondo et al., 2015*).

## Recombinant protein purification

All proteins were expressed in *E. coli* BL21 (DE3) cells. Cell cultures were grown at 37°C in LB liquid media supplemented with 50 µg mL$^{-1}$ kanamycin to an OD$_{600}$ of 0.6–0.8 before expression was induced by the addition of IPTG to a final concentration of 0.4 mM IPTG. The cultures were incubated at 20°C overnight before the cells from 2 L of cell culture were harvested by centrifugation at 3500 x g for 30 min. The cell pellets were resuspended in 70 mL of buffer A (50 mM Tris.HCl, pH 8, 300 mM NaCl, 10 mM imidazole) with 500 Kunitz units of DNase I and 1 mL Roche complete protease inhibitor cocktail at 25x working concentration. The cells were lysed by sonication, centrifuged at 19000 x g for 20 min and the supernatant was filtered using a 0.45 µm filter. The clarified cell lysate was loaded onto a 5 mL Ni-NTA superflow cartridge (Qiagen), washed with buffer A, and bound proteins were eluted with 50 mM Tris.HCl, pH 8, 300 mM NaCl, 250 mM imidazole. The His$_6$-tag of PrkA-KD was cleaved with His-tagged TEV protease (1 mg TEV for 20 mg of protein) at 4°C during an overnight dialysis against a buffer of 50 mM Tris.HCl, pH 8, 300 mM NaCl, 10 mM imidazole, 1 mM DTT; TEV cleavage of ReoM was conducted as above except the dialysis was carried out at 20°C. The proteolysis reaction products were then passed over a 5 mL Ni-NTA superflow cartridge (Qiagen) to remove TEV and uncleaved protein. The proteins that did not bind to the Ni-NTA column were concentrated and loaded onto either a Superdex 75 XK16/60 (GE Healthcare) (ReoM) or a Superdex 200 XK16/60 (GE Healthcare) (PrkA-KD and PrpC) equilibrated with 10 mM Na-HEPES, pH 8, 100 mM NaCl for size exclusion chromatography. Fractions from the gel filtration were analysed for purity by SDS-PAGE, concentrated to 20–40 mg mL$^{-1}$, and small aliquots were snap-frozen in liquid nitrogen for storage at −80°C.

## X-ray crystallography and ReoM structure determination

For ReoM, 23 mg mL$^{-1}$ of protein in 10 mM Na-HEPES pH 8, 100 mM NaCl was subjected to crystallisation by sparse matrix screening using a panel of commercial crystallisation screens. 100 and 200 nL drops of protein and 100 nL of screen solution were dispensed into 96 well MRC crystallisation plates (Molecular Dimensions) by a Mosquito (TTP Labtech) liquid handling robot and the crystallisation plates were stored at a constant temperature of 20°C. The crystals that grew and were subsequently used for diffraction experiments were formed in 0.1 M phosphate/citrate pH 4.2, 0.2 M lithium sulfate, 20 % w/v PEG 1000 from the JCSG + screen and were mounted onto rayon loops directly from the crystallisation drops and cryo-cooled in liquid nitrogen.

Diffraction data were collected on beamline I03 at the Diamond Light Source (DLS) synchrotron. Diffraction images were integrated in MOSFLM (*Battye et al., 2011*) and scaled and merged with AIMLESS (*Evans and Murshudov, 2013*). The initial model was generated by molecular replacement in PHASER (*McCoy et al., 2007*) using the dimeric, 20-conformer ensemble model (PDBid 5US5) of IreB solved by nuclear magnetic resonance (*Hall et al., 2017*) as a search model. The final model was produced by iterative cycles of model building in COOT (*Emsley et al., 2010*) with refinement in REFMAC (*Murshudov et al., 1997*) until convergence. The diffraction data collection and model refinement statistics are summarised in *Table 1*.

## Protein phosphorylation and dephosphorylation

The effect of phosphorylation and dephosphorylation on ReoM and PrkA-KD proteins was analysed by 20% non-denaturing PAGE. Phosphorylation reactions consisted of 18.5 µM ReoM, 3.7 µM PrkA-KD, 5 mM ATP and 5 mM MgCl$_2$, diluted in 10 mM HEPES.HCl pH 8.0 and 100 mM NaCl. Dephosphorylation reactions consisted of 37 µM P-ReoM, 3.7 µM PrkA-KD, 18.5 µM PrpC and 1 mM MnCl$_2$, diluted in 10 mM HEPES.HCl pH 8.0 and 100 mM NaCl. In each case controls were loaded at the same concentrations. The reactions were incubated at 37°C for 20 min prior to electrophoresis at 200 V for 2.5 hr on ice.

## Isolation of phosphorylated ReoM

Phosphorylation reactions consisted of 37 µM ReoM, 3.7 µM PrkA-KD, 5 mM ATP and 5 mM MgCl$_2$, diluted in 10 mM HEPES.HCl pH 8.0 and 100 mM NaCl, to a total volume of 5 mL. The protein mix was loaded onto a PD 10 desalting column to remove excess ATP and protein fractions were loaded onto a MonoQ 5/50 GL column. Buffer A consisted of 10 mM HEPES.HCl pH 8.0 and 100 mM NaCl

and buffer B was 10 mM HEPES.HCl pH 8.0 and 1M NaCl. Bound proteins were eluted over 25 mL with a 15–35% gradient of buffer B.

## Liquid chromatography-mass spectrometry

All liquid chromatography-mass spectrometry (LC-MS) analyses were performed using an Agilent 6530 Q-TOF instrument with electrospray ionisation (ESI) in positive ion mode, coupled to an Agilent 1260 Infinity II LC system, utilising mobile phase of 0.1% (v/v) formic acid in LC-MS grade water (A) and acetonitrile (B). Prior to peptide mapping, 10 µL of purified proteins (~1 mg/ml) were digested using Smart Digest Soluble Trypsin Kit (Thermo Fisher Scientific) according to the manufacturer's guidelines. Tryptic peptides and intact protein samples were extracted using HyperSep Spin Tip SPE C18 and C8 tips, respectively (ThermoFisher Scientific) before analysis. For phosphosite analysis, 10 µL of digest was injected onto an Acclaim RSLC 120 C18 column (Thermo Fisher Scientific, 2.1 × 100 mm, 2.2 µm, 120 Å) for reversed phase separation at 60°C and 0.4 ml/min, over a linear gradient of 5–40% B over 25 min, 40–90% B over 8 min followed by equilibration at 5% B for 7 min. ESI source conditions were nebuliser pressure of 45 psig, drying gas flow of 5 L/min and gas temperature of 325°C. Sheath gas temperature of 275°C and gas flow of 12 L/min, capillary voltage of 4000V and nozzle voltage of 300V were also applied. Mass spectra were acquired using MassHunter Acquisition software (version B.08.00) over the 100–3000 m/z range, at a rate of 5 spectra/s and 200 ms/spectrum, using standard mass range mode (3200 m/z) with extended dynamic range (2 GHz) and collection of both centroid and profile data. MS/MS fragmentation spectra were acquired over the 100–3000 m/z range, at a rate of 3 spectra/s and 333.3 ms/spectrum. For intact protein analysis,10 µL of desalted protein (~1 mg/ml) was injected onto a Zorbax 300 Å Stable Bond C8 column (Agilent Technologies, 4.6 × 50 mm, 3.5 µM) for reversed phase separation at 60°C and 0.4 mL/min, over a linear gradient of 15–75% B over 14 min, 75–100% B over 11 min followed by post-run equilibration at 15% B for 10 min. ESI source conditions were nebuliser pressure of 45 psig, drying gas flow of 5 L/min and source gas temperature of 325°C were applied. Sheath gas temperature of 400°C and gas flow of 11 L/min, capillary voltage of 3500V and nozzle voltage of 2000V were also used. Mass spectra were acquired using MassHunter Acquisition software (version B.08.00) over a mass range of 100–3000 m/z, at a rate of 1 spectra/s and 1000 ms/spectrum in extended mass range (20000 m/z) at 1 GHz. Acquired MS and MS/MS spectra were analysed using Agilent MassHunter BioConfirm software (version B.10.00) for identification of phosphorylated residues and subsequent intact mass determination with processing of raw data using maximum entropy deconvolution.

## Analytical size exclusion chromatography

Purified ReoM and P-ReoM proteins were run individually on a Superdex 200 Increase 10/300 GL column. 100 µl samples at 1.5 mg/mL were injected onto a column equilibrated in 10 mM HEPES.HCl pH 8.0 and 100 mM NaCl, with a flow of 0.75 mL/min.

## Acknowledgements

This work was funded by DFG grants HA 6830/1–1 and HA 6830/1–2 and a grant of the Fonds der Chemischen Industrie to SH. ZR was funded by a UK BBSRC DTP studentship to RJL (BB/M011186/1). We acknowledge Diamond Light Source for time on beamline I03 under proposal MX-18598 and Dr. Arnaud Basle for help with X-ray data collection. We thank Ulrich Nübel (Braunschweig) and Janina Döhling for help with some experiments and Petra Kaiser for technical assistance. We would like to thank the National BioResource Project (NIG, Japan): *B. subtilis* for sharing *B. subtilis* mutant strains. The co-ordinates and structure factors for the crystal structure of ReoM have been deposited at PDBe with accession code 6TIF.

## Additional information

### Funding

| Funder | Grant reference number | Author |
| --- | --- | --- |
| Deutsche Forschungsge-meinschaft | HA 6830/1-1 | Sven Halbedel |

| | | |
|---|---|---|
| Deutsche Forschungsge-meinschaft | HA 6830/1-2 | Sven Halbedel |
| Verband der Chemischen In-dustrie | 661460 | Sven Halbedel |
| Biotechnology and Biological Sciences Research Council | BB/M011186/1 | Richard J Lewis |

The funders had no role in study design, data collection and interpretation, or the decision to submit the work for publication.

### Author contributions

Sabrina Wamp, Zoe J Rutter, Jeanine Rismondo, Claire E Jennings, Conceptualization, Formal analysis, Investigation, Visualization; Lars Möller, Formal analysis, Investigation, Visualization; Richard J Lewis, Conceptualization, Supervision, Funding acquisition, Investigation, Visualization, Writing - original draft, Project administration; Sven Halbedel, Conceptualization, Formal analysis, Supervision, Funding acquisition, Investigation, Writing - original draft, Project administration

### Author ORCIDs

Sven Halbedel (iD) https://orcid.org/0000-0002-5575-8973

### Decision letter and Author response

Decision letter https://doi.org/10.7554/eLife.56048.sa1
Author response https://doi.org/10.7554/eLife.56048.sa2

## Additional files

### Supplementary files

- Supplementary file 1. Key resources table.
- Transparent reporting form

### Data availability

Genome sequences of shg8, shg10, shg12 and LMSW76 were deposited at ENA under study number PRJEB35110 and sample accession numbers ERS3927571 (SAMEA6127277), ERS3927572 (SAMEA6127278), ERS3927573 (SAMEA6127279), and ERS3967687 (SAMEA6167687) respectively. The co-ordinates and structure factors for the crystal structure of ReoM have been deposited at PDBe with accession code 6TIF.

The following datasets were generated:

| Author(s) | Year | Dataset title | Dataset URL | Database and Identifier |
|---|---|---|---|---|
| Wamp S, Rutter ZJ, Rismondo J, Jennings CE, Möller L, Lewis RJ, Halbedel S | 2020 | PrkA controls peptidoglycan biosynthesis through the essential phosphorylation of ReoM | https://www.ebi.ac.uk/ena/data/search?query=PRJEB35110 | European Nucleotide Archive, PRJEB35110 |
| Wamp S, Rutter ZJ, Rismondo J, Jennings CE, Möller L, Lewis RJ, Halbedel S | 2020 | PrkA controls peptidoglycan biosynthesis through the essential phosphorylation of ReoM | https://www.ebi.ac.uk/pdbe/entry/pdb/6tif | Protein Data Bank, 6TIF |

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
