## [Decision Letter]

**Acceptance summary:**

Peptidoglycan is an integral layer of bacterial cell walls and its biosynthesis and remodelling has been the target of antimicrobial chemotherapy for decades. Despite this longstanding history, there is much to be learnt about how synthesis and crosslinking of the peptidoglycan polymer is regulated in different bacterial species. Your study reports the regulation of peptidoglycan precursor metabolism through the activity of a Serine Threonine Protein Kinase, which uses a phospho-relay mechanism to modulate the stability of key peptidoglycan biosynthetic enzymes. The regulatory circuit described may have important implications for the study of cell wall homeostasis in other bacteria and will most likely generate new hypotheses.

**Decision letter after peer review:**

Thank you for submitting your article "PrkA controls peptidoglycan biosynthesis through the essential phosphorylation of ReoM" for consideration by *eLife*. Your article has been reviewed by three peer reviewers, one of whom is a member of our Board of Reviewing Editors, and the evaluation has been overseen by Michael Marletta as the Senior Editor. The following individuals involved in review of your submission have agreed to reveal their identity: Christoph Grundner (Reviewer #2); John Demian Sauer (Reviewer #3).

The reviewers have discussed the reviews with one another and the Reviewing Editor has drafted this decision to help you prepare a revised submission. In recognition of the fact that revisions may take longer than the two months we typically allow, until the research enterprise restarts in full, we will give authors as much time as they need to submit revised manuscripts.

Summary:

It has been demonstrated that a gpsB mutant of *Listeria monocytogenes* has defective peptidoglycan (PG) biosynthesis, together with a temperature sensitive defect, which can be suppressed by mutation of ClpC – a subunit of the ClpCP protease. Deletion of the protease results in the accumulation of enzymes involved in PG biosynthesis in *L. monocytogenes*, suggesting that proteolytic degradation is an important mechanism for regulating PG metabolism. In this study, you further study the gpsB mutant of *Listeria* and isolate suppressor mutants that allow for further description of the mechanism through which the activity of a Serine Threonine Protein Kinase controls PG precursor biosynthesis.

Key findings:

1) The authors isolate a further three suppressor mutants in the gps defective mutant of *L. monocytogenes*, in two cases mutations mapped to the *reoY* gene and in the third case, the mutation mapped to the RBS of the *reoM* gene, which the authors mutate to confirm the association with suppression of the temperature sensitive defect in the gps mutant. Expression of ReoM in the mutant background, restored the growth defect.

2) In the absence of ReoM or ReoY, the PG precursor synthesis enzyme, MurA, accumulated. Similar effects were seen when clpC was deleted individually or in combination with *reoM* or reoY, suggesting that *reoM* and *reoY* appear to control MurA levels in a ClpC dependent manner. Deletion of *reoM* and *reoY* homologues in *B. subtilis* led to similar accumulation of MurA.

3) The authors suggest that ReoM interacts with the PrkA kinase domain (KD) and PrkP phosphatase, when this interaction is tested in the presence of ATP, phosphorylated ReoM (at Thr7) is generated. It was also demonstrated that PrkP dephosphorylates phosphorylated ReoM. Expression of a phospho-ablative mutant of ReoM, in the corresponding deletion mutant, resulted in reduced viability, presumably due to reduced levels of MurA. This lethality was suppressed by deletion of clpC, suggesting that ClpC mediated degradation of MurA was responsibility for lethality seen upon expression of a phospho-ablative form of ReoM.

4) By solving the crystal structure of ReoM, the authors demonstrate that the Thr7 residues appears to be in a flexible region. The serendipitous presence of a sulphate ion in one of the protomers leads to a plausible idea about the function of phosphorylation that is tested by and consistent with mutagenesis.

5) By generating merodiploids with inducible copies of PrkA and PrkP, the authors demonstrate that these genes are essential and inversely regulate MurA levels. Deletion of ClpC (which could result in accumulation of MurA) alleviated the essentiality of PrkA.

Conclusions: Phosphorylation of ReoM is essential to control ClpCP-dependent proteolytic degradation of MurA. ReoY appears to be novel factor also required for degradation of ClpCP substrates.

Essential revisions:

1) Phosphoablation can have unintended consequences. A phosphomimetic ReoM mutant would be useful to strengthen the link to phosphorylation. A phosphomimetic that leads to MurA accumulation similarly to that seen with *reoM* deletion (and tolerates prkA deletion) would provide strong evidence for the model. Were such experiments attempted? If not, they should be considered to strengthen the conclusions.

2) To support the conclusions, analysis PG precursor levels appear necessary in this case. The entire model is based on protein stability but the functional consequences of this on cell viability and PG homeostasis have not been studied here. The relationship with resistance to cell wall targeting antibiotics is anecdotal. Perhaps assessing the thickness of the cell wall would provide more supporting information.

---

## [Author Response]

Essential revisions:1) Phosphoablation can have unintended consequences. A phosphomimetic ReoM mutant would be useful to strengthen the link to phosphorylation. A phosphomimetic that leads to MurA accumulation similarly to that seen with reoM deletion (and tolerates prkA deletion) would provide strong evidence for the model. Were such experiments attempted? If not, they should be considered to strengthen the conclusions.

We have engineered and analysed a phospho-mimetic *reoM T7D* mutant as requested by the reviewers. However, the *reoM T7D* mutant rather behaves as a non-phosphorylatable mutant (similar to *reoM T7A*) than as a real phospho-mimetic. We have included these results in Figure 5. Furthermore, we mention that phospho-mimetic mutations are not always successful (due differences in total charge and ionic shell size between phosphate and Glu/Asp residues, Dephoure et al. 2013, PMID:23447708) and refer to a similar effect observed with phospho-ablative and -mimetic Thr7 mutations in *E. faecalis* IreB (Hall et al. 2013, PMID:24080657).

2) To support the conclusions, analysis PG precursor levels appear necessary in this case. The entire model is based on protein stability but the functional consequences of this on cell viability and PG homeostasis have not been studied here. The relationship with resistance to cell wall targeting antibiotics is anecdotal. Perhaps assessing the thickness of the cell wall would provide more supporting information.

We thank the reviewers for this helpful point and have added electron microscopy analyses showing that *reoM* and *reoY* mutants have thicker PG layers at their cell poles and a ruffled PG layer along their lateral wall (Figure 3B). Furthermore, we no show that *reoM* and *reoY* mutants are salt sensitive and thus show a Δ*clpC*-like phenotype (Figure 3C). We hope that together with these additions and the reported ceftriaxone sensitivities, the reviewer agree that *reoM* and *reoY* deletions have functional consequences on growth and physiology of *L. monocytogenes*.

Despite these considerations, we felt the need to demonstrate that the *reoY* and *murZ* phenotypes can be complemented (a missing piece of information that had not been included in the previous version). We therefore included an experiment showing that reintroduction of *reoY* or *murZ* into their respective mutant backgrounds prevented MurA accumulation (Figure 2—figure supplement 1A).